# A random forest derived 35-year snow phenology record reveals climate trends in the Yukon River Basin

Caleb G. Pan[1], Kristofer Lasko[1], Sean P. Griffin[1], John S. Kimball[2], Jinyang Du[2], Tate G. Meehan[3], and Peter B. Kirchner[2]

[1]Geospatial Research Laboratory, Engineer Research and Development Center, US Army Corp of Engineers, Alexandria, VA, 22315, USA

[2]Numerical Terradynamic Simulations Group, W.A. Franke College of Forestry & Conservation, University of Montana, Missoula, MT, 59801, USA

[3]Cold Regions Research Engineering Laboratory, Engineer Research and Development Center, US Army Corp of Engineers, Hanover, NH, 03755, USA

*Correspondence to*: Caleb G. Pan (caleb.g.pan@erdc.dren.mil)

**Abstract.** This study presents a 35-year snow phenology record for the Yukon River Basin (YRB), developed using a Random Forest (RF) model at a 3.125 km resolution, capturing detailed trends in snowmelt onset and snowoff. The RF model, incorporating dynamic daily variables, improves upon traditional threshold-based methods by reducing sensitivity to transient thaw events and atmospheric noise. Model evaluation against station observations yielded a mean absolute error (MAE) of 10.5 days and a root mean square error (RMSE) of 13.7 days for snowmelt onset. For snowoff, the model achieved a MAE of 18.1 days and an RMSE of 20.7 days. This approach successfully mapped snow phenology across the diverse YRB landscape, providing valuable insight into how variations in snow cover align with regional climate patterns. Challenges such as sample bias due to limited ground-based data coverage highlight the need for expanding in situ measurements, to improve model performance further. Trend analysis segmented by two timeframes, 1988–2005 and 2006–2023, revealed distinct climate impacts on snow phenology. During 1988–2005, high snowfall and stable temperatures resulted in hastened snowmelt onset and lengthened snowmelt durations, reflecting early-season snow abundance. In contrast, from 2006–2023, warming spring and summer temperatures corresponded with progressively earlier snowmelt onset and snowoff. These shifts in snowmelt patterns align with a lengthened snow-free season, indicating increasing influence of warmer temperatures on the snowpack. This RF-derived dataset provides an essential tool for tracking climate-driven snow changes, offering insights into hydrologic and ecologic dynamics in the YRB under accelerating climate change.

## 1 Introduction

Snow cover and its seasonal progression, or phenology, play a crucial role in regulating the global energy budget and shaping ecosystem structure and function (Callaghan et al., 2011). These processes directly drive ecologic and hydrologic responses to seasonal variability. In the Yukon River Basin (YRB), regional warming (Ballinger et al., 2023; Rantanen et al., 2022) has reduced snow cover (Derksen and Brown, 2012), triggering widespread environmental changes. A warmer and longer snow-free season has disrupted permafrost, boosted vegetation growth, and increased ecosystem carbon uptake (Ling and Zhang, 2003; Pulliainen et al., 2017), but also enhanced regional drought and fire disturbance (Scholten et al., 2021) and led to a decline in plant diversity (Niittynen et al., 2018) and disrupted wildlife movements (Berger et al., 2018; Cosgrove et al., 2021). Seasonal snowmelt drives much of the discharge into the Yukon River and its adjoining stream networks, and the timing of this melt has significant hydrologic impacts. Earlier snowmelt has heightened flood risks, intensified the spring flood pulse,

and accelerated river ice breakup (Beltaos and Prowse, 2009; Lesack et al., 2014; Semmens and Ramage, 2013). These changes are reshaping the region's geomorphology and directly affecting the communities that rely on stable snow and ice conditions in the Yukon for winter travel, recreation, and harvest (Cold et al., 2020).

Enhanced monitoring and understanding in spatiotemporal variability in snow phenology are essential for assessing risks and mitigating potential impacts on Alaskan communities reliant on the Yukon River. Ground-based observations, like snow water equivalent (SWE) and snow depth measurements from SNOTEL sites, provide valuable insights into snow phenology. However, the vast landscape heterogeneity and limited ground observation locations make it challenging to forecast snow phenology reliably across large spatial scales (Bair et al., 2023). Satellite microwave remote sensing offers a valuable alternative for mapping snow phenology, especially in remote, high-latitude regions. The moderate frequency ($\sim\leq$37 GHz) retrievals from operational satellite microwave radiometers are sensitive to snow cover conditions, providing nearly continuous, year-round data. Importantly, the propagation of microwave energy through the snowpack is responsive to changes in snow structure, including liquid water content (LWC), grain size and density, which are key indicators of snowmelt onset (Tedesco et al., 2015). However, the sampling footprint from the passive microwave (PMW) retrievals can range from ~12-25 km resolution depending on frequency and can be too coarse to capture snow spatial heterogeneity, especially in mountain environments. While higher-frequency K-band and Ka-band radiometers are limited by their coarser spatial resolution, they provide twice-daily acquisitions for polar latitudes from 1988 to the present, offering a valuable long-term data record

In contrast, Synthetic Aperture Radar (SAR) sensors are sensitive to snow conditions and offer improved spatial resolution over microwave radiometers and scatterometers. C-band SAR data from the European Space Agency (ESA) Sentinel-1 mission has proven valuable for detecting snowmelt onset using a median minima backscatter approach, often in combination with optical-infrared remote sensing imagery (Darychuk et al., 2023; Gagliano et al., 2023; Marin et al., 2020; Nagler and Rott, 2000). The ability of SAR to detect changes in snowpack structure and LWC makes it particularly effective for identifying the onset of snowmelt, as the C-band radar backscatter at VV and VH polarizations decreases when snow transitions from dry to wet. The extraction of snowmelt onset using Sentinel-1 missions shows great promise, providing excellent detail with a spatial resolution of 10 meters. However, a current limitation of these data is the relatively short temporal record. Sentinel-1A began operations in April 2014, followed by Sentinel-1B nearly two years later in April 2016. Unfortunately, Sentinel-1B was decommissioned in December 2021 due to power issues.

Several snow phenology algorithms utilize K- and Ka-band radiometric brightness temperature (Tb) measurements collected from the Defense Meteorological Satellite Program (DMSP) Special Sensor Microwave Imager (SSM/I) (1987-present) and Special Sensor Microwave Imager/Sounder (SSMIS) (2004-present). Various retrieval algorithms using these data to derive snow properties include: 1) the Tb diurnal amplitude variation (DAV) method (Ramage and Isacks, 2002; Tedesco and Miller, 2007), 2) the Tb differencing approach (K-Ka) (Wang et al., 2013, 2016), 3) the use of a single frequency Tb temporal change

algorithm coupled with reanalysis surface temperature (Kim et al., 2017), 4) the gradient ratio polarization (GRP) approach (Dolant et al., 2016; Du et al., 2025; Pan et al., 2018) and 5) a remote sensing and physics-based hybrid method (Dattler et al., 2024). Each algorithm leverages the interaction between the surface snowpack, its liquid water content (LWC), and the resulting effect on the Tb signal at each band or polarization. Specifically, dry snow conditions lead to volumetric scattering in both K and Ka bands, with stronger scattering at higher frequencies. In contrast, when the LWC within the snowpack increases, snow emissivity at lower frequencies likely decreases due to attenuated soil emission, while increasing at higher frequencies such as K and Ka bands due to enhanced emission from the wet snow layers (Dolant et al., 2016). Due to these interactions, past algorithms have successfully derived snow phenology by analyzing Tb timeseries using these approaches and applying thresholds to identify transitioning snow conditions.

While threshold-based methods have successfully predicted snow phenology, they often fail to fully capture landscape variability in snow conditions due to their coarse spatial resolution. Additionally, these methods are susceptible to atmospheric noise, which can lead to potential false positives. Alternatively, machine learning (ML) offers a flexible empirical modeling approach for estimating snow properties from satellite observations and other ancillary data. ML provides the ability to model complex interactions across diverse datasets and has been applied widely in cryosphere applications (Campbell et al., 2021; Dunmire et al., 2024; Guidicelli et al., 2023; Meloche et al., 2022; Tedesco et al., 2004; Tsai et al., 2019). Among ML methods, random forest (RF) has demonstrated success, often bettering other methods, due to its flexibility, ability to handle high-dimensional data, and success in handling complex environmental datasets. RF constructs multiple decision trees during training and aggregates their outputs, reducing overfitting and increasing robustness in diverse datasets. Furthermore, RF can manage missing data and maintain accuracy even with uncorrelated features (Breiman, 2001). Neural networks (NN) also present a strong alternative, particularly for modelling nonlinear and temporal relationships in snow related data. NN methods have shown strong performance in linking PMW observations to snow properties (Forman and and Xue, 2017). However, model comparisons have shown that RF has outperformed NN for retrieving snow cover, emphasizing RF's robustness in cryosphere remote sensing applications (Xiao et al., 2021).

Previous efforts to characterize long-term snow phenology in the YRB have relied on threshold-based methods applied to PMW observations (Pan et al., 2021; Semmens et al., 2013). While these approaches have provided valuable insights, they face limitations related to model generalizability, temporal robustness, and sensitivity to landscape heterogeneity. Fixed thresholds can misclassify short-term warming events as melt onset and are often unable to capture the spatial complexity of snowpack transitions in mountainous terrain. In this study, we address these challenges by applying a ML framework that incorporates dynamic predictors and static landscape features. This approach enables daily classification of snow state, spatially explicit uncertainty mapping, and improved representation of the nonlinear interactions that govern snowmelt.

Furthermore, our dataset extends the snow phenology records for the YRB from 2018 to 2023, offering a 25% increase in temporal coverage during a period of record-setting Arctic warming (Ballinger et al., 2023). The extended study period enables us to examine emerging trends in the snow season which may have been missed in prior datasets.

Our study integrates a temporal component into the RF framework, allowing the model to capture seasonal variations in snow cover. Unlike traditional thresholding methods that rely on fixed values (Pan et al., 2021), the RF model accounts for multiple variables and their interactions, producing more nuanced predictions. By incorporating timeseries data, our RF model tracks the evolution of snow conditions throughout the season (Rittger et al., 2021), improving predictions of snowmelt onset.

In this paper, we examine the question: How amplified Arctic warming has influenced the timing, duration, and variability in

snow phenology in the YRB? To address this question, we use an ML framework informed with Tb timeseries from the K- and Ka-bands collected from SSM/I(S), along with other complimentary dynamic and static variables, to estimate primary spring snowmelt onset and snowoff dates across the YRB from 1988 to 2023. The resulting annual snow phenology maps are produced at an enhanced resolution of 3.125 km, offering an improvement over previous records derived directly from PMW observations and enabling a more detailed delineation of landscape heterogeneity. We then apply the snow phenology outputs

with other ancillary and in-situ environmental data to: 1) assess model performance and define relative quality maps, 2) examine YRB snow phenology climatology and compare it with anomalous years, 3) analyze spatiotemporal trends in snow phenology over the period of record, and 4) explore interactions between snow phenology and seasonal snowfall and temperature trends.

## 2 Study Area

The YRB constitutes one of North America's largest river basins (Figure 1). This region experiences six to nine months of snow cover annually, and spring snowmelt runoff is the main hydrologic contribution to the discharge (Brown et al., 2020). The YRB has a mean annual discharge of 6400 m$^3$ s$^{-1}$ (Brabets et al., 2000), with a drainage area exceeding 853,300 km$^2$ and covers 10 degrees of latitude from 59°N to 69°N, extends into the Canadian Yukon and British Columbia territories to the east, and the west coast of Alaska before draining into the Bering Sea. The diverse topography, with a median elevation of 617 m

and extending from sea level to the highest elevations of the Brooks (2735 m) and Alaska (6190 m) Ranges, encompasses a diversity of northern boreal, arctic, alpine and maritime biomes. Evergreen needleleaf forests are the dominant vegetation cover (54%) followed by broadleaf deciduous forests (9%) covering the valley bottoms and into the mid-elevations. The Yukon Delta and higher elevations have tall and low shrubs (9%) mixed with some dry and wet herbaceous (9%) tundra as the dominant plant community. Permafrost is present to a large extent in the YRB, and comprises several types including sporadic

(14%), discontinuous (46%) and continuous (16%) and moderately thick to thin permafrost (24%) (Brabets et al., 2000). Historically the Yukon River served as the main travel corridor of the region and theYRB is the ancestral homelands of several

Native Alaskan culture. Presently, many communities are inextricably linked to and rely upon the Yukon and its tributaries for travel, subsistence, and livelihood (Cold et al., 2020).

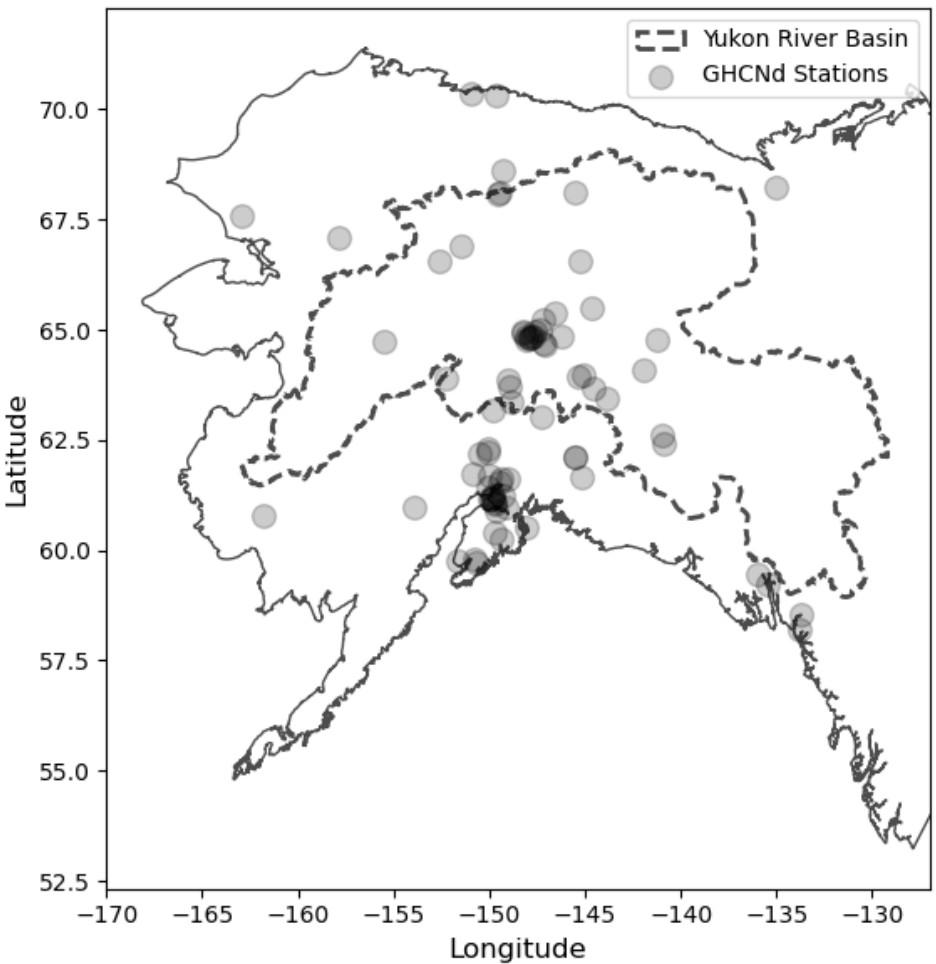


**Figure 1: The YRB study areais delineated by the dashed black polygon and the black upside-down triangles indicate climate stations utilized for creating a training and testing dataset for our ML models.**

## 3 Data

### 3.1 Training and Testing Datasets

We acquired daily in-situ snow depth measurements from the Global Historical Climatology Network (GHCNd) (Menne et al., 2012) to build the RF model training and testing dataset. Filtering stations across Alaska, 77 stations included snow depth measurements spanning at least one year between 1988 and 2023. While our focus is on the YRB, we included sites located

outside the basin to supplement the sparse in-situ observations within the YRB and enhance model generalizability. This expanded dataset increases the robustness of the model across diverse snowmelt regimes. However, we acknowledge that incorporating data from regions with different climatic conditions (e.g., maritime vs. continental) may introduce some bias, and we interpret model outputs within the YRB with that potential limitation in mind.

Although many researchers use the day of peak SWE or a breakpoint after peak SWE to determine the onset of snowmelt (Darychuk et al., 2023; Gagliano et al., 2023), the lack of SWE measurements in the GHCNd led us to use peak snow depth instead. Specifically, for each station and year, we identified snowmelt onset by locating the day with the highest snow depth in spring (Mar-May). However, peak snow depth often did not accurately represent the true onset of snowmelt, as decreases in snow depth can occur due to factors like wind redistribution, sublimation, or compaction, unrelated to snowmelt. For this reason, we could not apply rigid rules to defining snowmelt onset using snow depth.

To improve the identification of snowmelt onset, we used our best judgment for interpretation, supported by air temperature data. When average daily temperatures consistently rose above freezing and snow depth began to steadily decrease from its peak, it became easier to pinpoint the onset of snowmelt. Identifying snowoff from snow depth was more straightforward and defined as the first day when snow depth reached 0 for at least 10 consecutive days in spring. After analyzing each in-situ snow depth timeseries, we compiled 971 observations for snowmelt onset and 933 snowoff observations for RF training and testing. The lower number of snowoff observations is primarily due to gaps in the snow depth records—many timeseries did not extend far enough into the melt season to capture when snow depth reached zero.

### 3.2 Timeseries Datasets

In this study, we employ a combination of dynamic and static datasets as RF model predictors and for analyzing the model snow phenology outputs. The dynamic RF predictors include the Tb-derived indices, Tb Difference (TBD) and Gradient Ratio Polarization (GRP), as well as their respective 3-day moving averages (MA_TBD and MA_GRP). We also utilize daily Thaw Degree Days (TDD), day of year (DOY), total water vapor (TQV), and daily snow cover. Together, these dynamic datasets provide a comprehensive basis for capturing both seasonal and interannual variability in snow phenology.

We also include several static landscape factors and assess how landscape features influence model sensitivity. Static variables include fractional water (FW) cover, fractional tree cover (FTC), elevation (GTOPO), topographic variability, aspect, and proximity, described by a pixel's proximity to the nearest ocean. These datasets are summarized in Table 1 and described in more detail in the following section. In addition, a comprehensive table with all datasets used in this study are found in Table A1.

**Table 1: Dynamic and static predictor summary, including their abbreviation, spatial and temporal resolutions.**

| Dataset | Spatial Resolution | Temporal Resolution |
| --- | --- | --- |
| Tb Difference (TBD) | 3.125 km | daily |
| Gradient Ratio Polarization (GRP) | 3.125 km | daily |
| Moving Average TBD (MA_TBD) | 3.125 km | 3-day moving average |
| Moving Average GRP (MA_GRP) | 3.125 km | 3-day moving average |
| Cumulative Thaw Degree Day (TDD) | 1 km | daily |
| Day of Year (DOY) | | daily |
| Snow cover | 4 km and 3 km | daily |
| Total Precipitable Water Vapor (TQV) | 50 km | daily |
| Fraction Water (FW) | 1 km | static |
| Fractional Tree Cover (FTC) | 250 m | static |
| Elevation (GTOPO) | 1 km | static |
| Aspect | 1 km | static |
| Proximity | 1 km | static |
| Topographic Variability | 30 m | static |

### 3.2.1 Passive Microwave Satellite Record

We acquired K-band (19 GHz) and Ka-band (37 GHz) afternoon Tb retrievals at vertical (V) and horizontal (H) polarizations from the MEaSUREs Calibrated Enhanced Resolution Passive Microwave Daily EASE-Grid 2.0 Brightness Temperature ESDR, available from the National Snow and Ice Data Center (NSIDC) (Brodzik and Long, 2016). This Tb record is

multidecadal and calibrated across multiple sensors and platforms from different frequencies and polarizations from the NOAA DMSP SSM/I and SSMIS. Each platform has several sensors, from SSMI/I we selected F08 (1998-1991), F11 (1992-1995), F13 (1996-2007) and from SSMIS we used F17 (2007-2016) and F18 (2017-2023). These sensors were selected because their equatorial overpass time remained consistent while in commission. Missing temporal observations were gap-filled using a temporal linear interpolation of adjacent Tb retrievals (Wang et al., 2016). These missing observations are generally infrequent

and short in duration, typically affecting only 1-2 consecutive days due to sensor outages, orbital gaps, or data transmission gaps (Long and Brodzik, 2016).

We chose to focus only on the SSM/I-SSMIS record to ensure continuity and minimize calibration uncertainty across the 35-year period. These sensors offer comparable radiometric characteristics and orbital parameters, enabling a harmonized

timeseries suitable for long-term analysis. Other PMW sensors such as SMMR, AMSR, and AMSR2 were not included. The SMMR dataset (1978-1987) exhibits frequent data dropouts, and its inclusion would have required extensive temporal interpolation. The Advanced Microwave Scanning Radiometer for EOS (AMSRE) and the Advanced Microwave Scanning Radiometer 2 (AMSR2) offer valuable observations, but their shorter and non-overlapping time periods with the SSM/I-SSMIS record would require additional cross-sensor calibration and harmonization to reconcile differences in spatial resolution and

observation geometry. By focusing on SSM/I-SSMIS, we preserved the temporal consistency and data integrity critical for reliable empirical model training and long-term phenological trend detection.

Native sampling resolution of the combined K and Ka Tb retrievals are ~25 km or coarser, however the MEaSUREs products used were processed using the scatterometer image reconstruction (SIR) approach to obtain an enhanced spatial grid resolution of 6.25 km (K) and 3.125 km (Ka) from the overlapping Tb antenna patterns (Brodzik et al., 2018; Long and Brodzik, 2016). We then resampled K-band Tb retrievals to match the Ka resolution of 3.125 km using a nearest neighbor interpolation. We then reduce the vertically polarized K and Ka bands into a Tb difference index, henceforth termed as TBD, defined as the difference between K and Ka bands (Wang et al., 2013). We also reduce the K and Ka bands into an additional index, the GRP by first calculating the Gradient Ratio (GR) at vertical and horizontal polarizations using equation 1 (Grenfell and Putkonen, 2008):

$$GR\left(pol_{(37,19)}\right) = \frac{[T_b\,(pol,37) - T_b\,(pol,19)]}{[T_b\,(pol,37) + T_b\,(pol,19)]} \tag{1}$$

The GRP is then ratioed using equation 2 (Dolant et al., 2016):

$$GRP = \frac{GR_V}{GR_H} \tag{2}$$

Together, both the TBD and GRP provide a source for identifying daily snow conditions such as dry and stable, melting, and disappeared(Pan et al., 2019; Wang et al., 2016).

### 3.2.2 Daily Snow Cover

We use ancillary daily snow covered area estimates to determine the presence or absence of snow at a given location and time. For the period from 1988 to 2023, we relied on two data sources. From 2004 to 2023, we used the Interactive Multisensor Snow and Ice Mapping System (IMS) daily snow cover extent record, which has a 4 km resolution. The IMS provides global coverage and is informed by expert interpretation of geostationary visible satellite imagery, polar-orbiting multispectral satellite sensors, PMW sensors, and ground observations (Helfrich et al., 2007).  Although the IMS also provides daily snow cover area outputs at a coarser 24 km resolution dating back to 1997, we opted to use alternative higher spatial resolution snow cover estimates from SnowModel (Liston et al., 2020) to fill in the earlier years (1988–2003), as the 24 km resolution IMS data is less able to resolve snow cover heterogeneity in complex terrain and does not cover the entire period of interest.

The SnowModel dataset was developed using meteorological data derived from both MERRA-2 and ERA5 reanalysis data. These reanalyses were used to create bias-corrected inputs for SnowModel, resulting in daily estimates of snow properties for the North American domain at a 3 km resolution from 1980 to 2020 (Liston et al., 2023, Liston et al., 2020). Although SnowModel includes several snow variables, we used the modeled snow depth to define daily snow presence or absence.

Specifically, if the estimated snow depth exceeded 0 on any given day, we assigned a value of 1; if snow depth was 0, we assigned a value of 0.

### 3.2.3 Daymet

We calculated daily cumulative thaw degree days (TDD) using the North American Daymet (V4) record (Thornton et al., 2021). We obtained the Daymet data through the Microsoft Planetary Computer STAC, which is produced by the Oak Ridge National Laboratory DAAC. Daymet provides 1 km spatial resolution, interpolated from daily weather station temperature observations, but with potential bias introduced from the sparse regional weather station network, especially at higher elevations. TDD serves as a useful proxy for assessing the amount of incoming solar radiation the snowpack has been exposed to at a given location and reflects the seasonal dynamics of anomalous temperatures that influence snowmelt onset. TDD was created by summing daily mean temperatures above 0 degrees Celsius for each pixel during the melt season. To facilitate comparison across regions and years, TDD values were normalized to a cumulative percent scale (0-100%), where 0% represents the onset of above-freezing temperatures and 100% represents the total seasonal accumulation. TDD was created by summing daily mean temperature above 0 degrees Celsius for each pixel and is returned as a cumulative percent.

### 3.2.4 MERRA-2 Total Column Water Vapor

Precipitable water vapor and precipitating clouds can affect PMW observations (Du et al., 2015) and assert adverse effects on snow retrievals (Dolant et al., 2016). To account for these effects on SSM/I and SSMIS evening observations, we incorporated total column water vapor (TQV) from the NASA MERRA-2 (Modern-Era Retrospective analysis for Research and Applications, Version 2) product (Gelaro et al., 2017). MERRA-2 is a global atmospheric reanalysis dataset that provides TQV estimates at a native spatial resolution of 0.5 degrees latitude x 0.625 degrees longitude and spans from 1980 to present. We extracted TQV values corresponding to 6 PM Alaska Local Time to coincide with the satellite overpasses and resampled to 3.125 km using a nearest neighbour interpolation.

### 3.3 Static Datasets

We also used several static datasets for model training and to examine the influence of landcover on snow phenology prediction. We represented elevation using the GTOPO30 dataset at a 1 km resolution and used it to derive terrain aspect. To better capture fine-scale terrain variability, we also incorporated additional topographic metrics derived from the ALOS World 3D-30m DEM (Tadono et al., 2014). Specifically, we calculated the standard deviation of elevation at the 3.125 km EASE-grid scale, which provides a proxy for local topographic complexity and helps mitigate known Tb retrieval uncertainties in high-relief regions, from henceforth this variable is referred to as topographic variability (Xiong et al., 2022).

We acquired average fractional water inundation (FW) from the global land parameter data record, generated from the AMSRE and AMSR-2 records (Du et al., 2017). In addition to prediction and assessing uncertainty, FW served as a mask to screen model outputs likely affected by water contamination.

To represent percent fractional tree cover (FTC), we utilized the MODIS MOD44B V005 500m Vegetation Continuous Fields product. Additionally, we created a custom dataset, termed 'proximity,' which captures the distance of each pixel from the ocean. This is important because Tb pixels near large water bodies are prone to water contamination and frequent cyclonic events that can influence LWC in the regional snowpack (Rees et al., 2010).

### 3.4 Ancillary Datasets

We used an established satellite-based snow phenology dataset for comparison with our ML results. These data include the annual timing of snowmelt onset and snowoff for the YRB from 1988 to 2018 mapped to a 6.25 km resolution grid as a day of year (DOY) (Pan et al., 2020, 2021). The data were also derived using a similar thresholding approach of the GRP and TBD derived from microwave Tb observations. Additionally, a glacier landcover dataset for the YRB was obtained from the Glacier Covered Area for the State of Alaska dataset (Roberts-Pierel et al., 2022), which we used to identify pixels that maintain year-

round snow cover, as they do not experience a 'snowoff.'

### 4 Methods

### 4.1 RF Framework

We implemented a RF classifier to predict snow phenology, specifically focusing on estimating the annual timing (DOY)of snowmelt onset and snowoff across the YRB. The RF approach was chosen due to its ability to handle complex, high-

dimensional data, and robustness to overfitting, making it well-suited for cryosphere applications (Breiman 2001, Alifu et al., 2020; Blandini et al., 2023). Although RF does not inherently model temporal sequences like some other algorithms, our approach used daily inputs to generate a sequence of predictions that are later interpreted in chronological order. Each day is treated as an independent sample during training and prediction, but the outputs are interpreted sequentially to identify the timing of snowpack transitions, such as the first wet snow day indicating snowmelt onset. This post-prediction sequencing is

critical for deriving the correct DOY phenology metrics. Accordingly, we used the RF implementation in scikit-learn (Pedregosa et al., 2011).

In this study, the RF model snow phenology metrics were derived at 3.125 km resolution from 1988-2023, representing a valuable spatial and temporal enhancement over previous snow records developed for the YRB (Pan et al., 2021). The earlier

threshold-based dataset was produced at 6.25 km and covered the years 1988-2016, later extended to 2018. In contrast, the

dataset presented here extends through 2023, offering a 7-year increase in temporal coverage. Although the original K-band Tb grid resolution was 6.25 km, the 3.125 km resolution of the RF predictions is consistent with the grid resolution of the Ka-band Tb record, which may help to improve the spatial delineation of snowpack characteristics. Additionally, the 3.125 km resolution is approximate to—or still coarser than—other RF model predictor datasets used, which helps to ensure spatial coherence in representing landscape heterogeneity.

### 4.1.1 RF Model Setup

To enhance the prediction of snow phenology, we configured the RF model to delineate daily snow conditions. We did this by classifying expected snow conditions for each day in a timeseries leading up to the observed snowmelt onset or snowoff day in spring as either 'dry snow' or 'present.' After the observed onset or snowoff day, the conditions are labeled as 'wet snow' or 'absent', respectively. The labeling approach transforms each timeseries into a sequence of daily snow condition classes, enabling a clear and daily description of a given evolving phenological event. By labeling the timeseries accordingly, we were able to: 1) add a temporal dimension to the RF classifier, 2) expand the RF training and testing datasets, and 3) use the day each labeled timeseries changes as the snow phenology date (Figure 2).

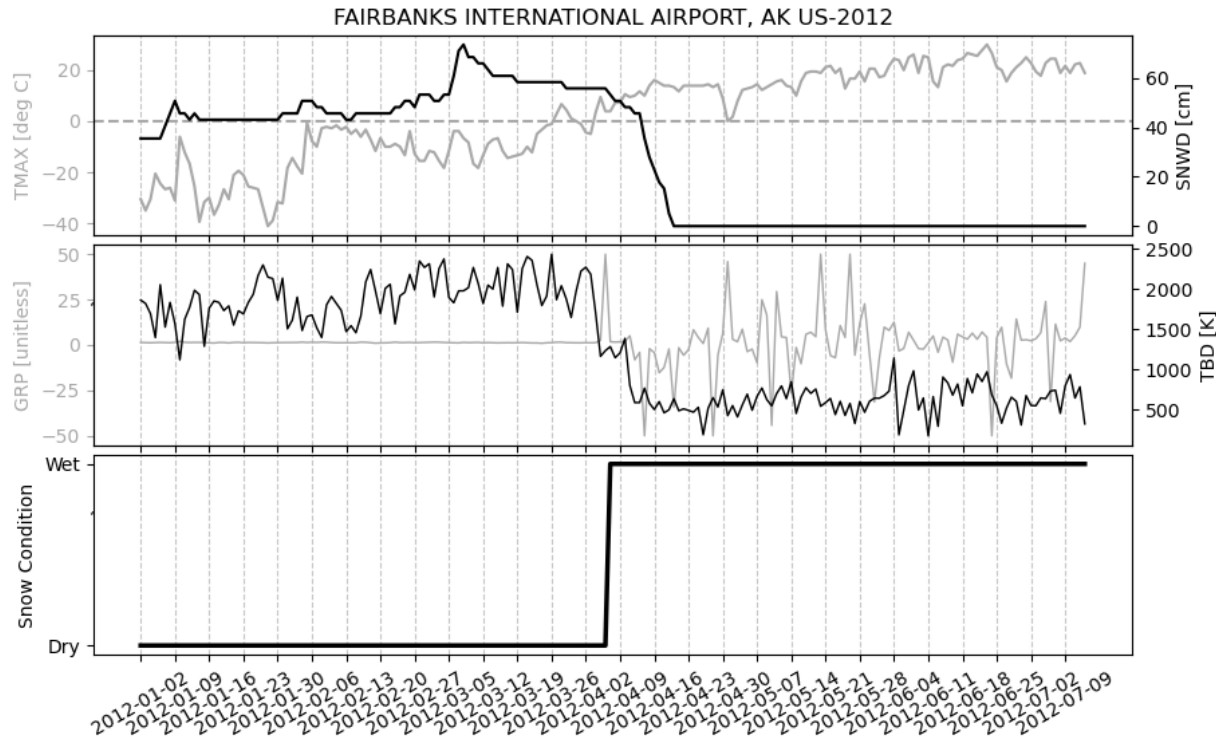

**Figure 2: Comparison between daily in situ air temperature and snow depth measurements (top plot) taken from Fairbanks International Airport in 2012 and collocated brightness temperature derived TBD and GRP (middle plot). The bottom plot shows the daily snowmelt onset RF model output. The snowmelt onset, marked by the transition from dry to wet snow, is identified as the day when the model output changes from 0 to 1.**

To assess daily snow conditions, each model was trained using a set of daily and static predictors. Daily predictors included TDD, TBD, GRP TQV, and DOY. Static predictors, representing landscape and environmental characteristics, included proximity to oceans, FTC, FW, elevation, topographic variability, and aspect. All static variables were scaled from 0 to 1. Table 1 includes more detail on the model training datasets. Snowoff included snowmelt onset as a predictor with the intention that this variable would ensure that snowoff predictions would occur after the snowmelt onset.


We parameterized the RF models using a cross-validated grid search method. This approach systematically evaluates various combinations of hyperparameters to identify the best configuration by performing cross-validation. It selects the combination of parameters that minimizes the user-defined evaluation metric, such as the cross-validated score (Jääskeläinen et al., 2022). The grid of adjustable parameters we provided for hyper-tuning included the number of the RF decision trees, the maximum

depth of each tree, the minimum sample size for node splitting, the minimum sample size for leaf nodes, and the maximum number of features considered at each split. In addition to parameter selection, the cross-validated score also helps minimize overfitting. A full list of tested parameter ranges and the selected final values for each model are provided in Table A2.

Finally, we identified the timing of snowmelt onset and snowoff from the model outputs by applying a logic that returned the

first day of 10 consecutive days classified as either 'wet snow' or 'absent.'

### 4.1.2 Assessing Uncertainty and Error

To ensure independence between training and testing data, we implemented our 80/20 split at the timeseries level. Such that, each timeseries represents a single pixel across a full snow season and is treated as an induvial unit. This precaution avoided any overlap in time or space between the training and testing sets. Consequently, no individual pixel's timeseries contributed

data to both training and testing subsets in each iteration, ensuring that model evaluation reflected only out-of-sample predictions.

Model performance was assessed using a bootstrapping approach, with an 80/20 split between training and testing data. For each iteration, both datasets were randomly sampled with replacement from the full dataset, allowing us to evaluate model

accuracy and variability. Performance was evaluated with the testing data by 1) extracting the $R^2$ value to quantify the agreement between observed and predicted dates, and 2) aggregating the Mean Absolute Error (MAE) across different landcover types. To determine whether differences in model error across landcover characteristics were statistically significant, we applied a one-way Analysis of Variance (ANOVA). For each iteration, we also calculated feature importance using the

built-in mean decrease in impurity method from the RF algorithm and calculated the average importance and standard deviation for each feature.

The output absolute error from our model bootstrapping was used as the dependent variable in an ordinary least squares (OLS) regression, with landcover variables such as FW, FTC, elevation, topographic variability, aspect, and proximity serving as the explanatory variables (Kim et al., 2011). The goal was to establish a relationship between the observed error and the landcover characteristics to identify pixels in the YRB where we may expect lower or higher errors. We then applied the OLS model across the YRB to predict anticipated error. These values were scaled from 0 to 1, creating a dimensionless quality control (QC) metric. The QC metric was further classified into a quantile classification, with qualitative labels of 'Best', 'Good,' 'Moderate,' and 'Low' to describe the relative quality of the model predictions. This qualitative classification simplifies interpretation and communication of model quality across the basin. While this process discretizes a continuous variable, the underlying QC values are retained and available.

## 4.2 Trend Analysis

### 4.2.1 Snow Phenology

To analyze snow phenology trends over time, we developed snow phenology climatology for the period 1991–2020, which corresponds to the current 30-year climate normal (Palecki et al., 2021). Using a natural break classification method, we divided the data into two categories: 'earlier' and 'later' snow events. In this context, a "snow event" refers to an annual snow phenology transition—such as snowmelt onset or complete snow disappearance—at the pixel level. Each pixel's event date was compared to its long-term climatological mean (1991–2020). If the event occurred earlier than the mean, it was classified as *"earlier"*; if later, it was classified as *"later."*

This classification enabled us to calculate the total area (in km²) of the basin experiencing earlier or later snow events in each year. This spatially aggregated approach provides a more hydrologically meaningful metric than raw day-of-year values, as it reflects how much of the landscape is contributing to earlier or delayed runoff. Framing the results relative to the climatological mean also contextualizes each year's snowmelt timing within a long-term baseline, helping to interpret the magnitude and direction of interannual variability.

Next, for each year, we classified the snow metrics using the same two categories derived from the climatology and calculated the annual change in area for each class. If the area of the *'earlier'* class decreased, we expected a corresponding increase in the *'later'* class. To assess the trends over time, we applied a linear regression and performed a Mann-Kendall Test (MKT) to evaluate the direction and strength of the annual changes in area, where the MKT outputs tau, a dimensionless, non-parametric measure of monotonic trend and strength (Kendall, 1948).

.

Because trends over the full period of record were generally weak or non-significant, we further examined the potential for meaningful sub-period patterns by dividing the 35-year timeseries (1988–2023) into two equal halves: 1988–2005 and 2006–2023. This split allowed us to evaluate whether changes in snow phenology were occurring within shorter timeframes that may

have been masked by variability across the full record.

### 4.2.2 Temperature and Snow Depth

Seasonal air temperature across the YRB was analyzed using data from GHCNd climate stations to assess long-term climate trends and their relationship to changes in snowpack conditions. To create a single, harmonized air temperature timeseries, we selected stations with at least 17 years of data for each of the two time periods (1988–2005 and 2006–2023) from an initial set

of 35 stations in the YRB. This selection criterion reduced the set to 8 stations for 1988–2005 and 15 stations for 2006–2023. With the selected stations, we then calculated seasonal average temperature timeseries for each period, specifically for winter, spring, summer, and combined spring/summer temperatures. These seasonal means were used in a trend analysis to evaluate how air temperatures have changed over time across the YRB and to explore their potential associations with observed snowmelt dynamics.


We also extracted the snow depth at the day of snowmelt onset across YRB using the GHCNd climate stations. Like temperature, we required a station to have recorded at least 17 years of snow depth measurements. We also checked each of these stations, to determine if the annual snow depth measurements were complete and without extended data gaps.. These screening criteria resulted in 4 stations selected for 1988-2005 and 13 stations for 2006-2023. The resulting snow depth data

were used to analyze trends in snow cover.

## 5 Results

### 5.1 RF Model Performance

The RF model effectively classified daily snow conditions for both snowmelt onset and snowoff, as demonstrated by the bootstrapped testing results. For snowmelt onset, the model classified snow conditions as either 'dry snow' or 'wet snow' with

high accuracy. The model achieved an F1-score, precision, and recall averaging 0.97 on the testing data, indicating a strong ability to balance false positives and false negatives. These metrics suggest that the RF model reliably distinguished between dry and wet snow conditions leading up to snowmelt onset. Gridsearch results for the RF are found in Table A2.

Similarly, for snowoff, the RF model successfully classified daily snow conditions as either 'present' or 'absent'. The model

maintained an average F1-score, precision, and recall of 0.96 on the testing data. This consistent performance highlights the

model's ability to accurately capture the transition between snow presence and absence throughout the snowoff period, providing dependable predictions of snow cover dynamics.

Once the snowmelt onset and snowoff days of year were extracted, they were compared against our testing data generated during bootstrap iterations. These stations are located both inside and outside the YRB. In each iteration, an 80/20 (training/testing) split with replacement ensured that the testing data represented a unique subset of high-quality observations from different years and locations, allowing the model's generalizability to be evaluated across a variety of conditions. The bootstrapped results yielded an average $R^2$ of 0.72 for snowmelt onset and 0.80 for snowoff, demonstrating that the predicted snow DOY metrics closely matched the observed values from the testing data. Additionally, the model produced a MAE of 6.11 [days] for snowmelt onset and 5.73 [days] for snowoff. The Root Mean Square Error (RMSE) values of the model results were 8.30 [days] for snowmelt onset and 7.70 [days] for snowoff. We also assessed model bias by evaluating the mean error across all testing observations. Results showed minimal bias in both snowmelt onset (mean error = 0.59 days) and snowoff timing (mean error = 0.20 days), indicating that the model does not systematically over or under predict the timing of these events. Overall, these metrics indicate favorable model performance in predicting the timing of these key snow phenology events across the YRB.

The final error assigned to the snow phenology dataset is assessed by comparing the RF model predictions across the full YRB gridded dataset with an independent testing dataset derived from the limited number of GHCNd stations located within the YRB. From these stations we calculated a MAE 10.54 [days] and RMSE of 13.68 [days] to the snowmelt onset product. The modeled snowoff results showed a MAE of 18.1 [days] and RMSE of 20.77 [days] relative to the station observations. The higher final observed errors, compared to the bootstrapped errors, are attributed to the greater heterogeneity in landcover and terrain across the basin, which contributes to larger differences between the sparse in-situ ground stations measurements of local snow conditions and the coarser landscape level RF predictions.

### 5.1.1 Model Feature Importance

On average, the most influential features for predicting snowmelt onset were DOY, TDD, TQV, snowcover presence and TBD, in that order (Figure A1). The snowoff predictions followed a similar pattern, with TDD, DOY, snow cover presence, and TBD emerging as the top-ranked features (Figure A2). In both models, the dynamic, timeseries features—such as temperature and snow cover presence—played a significantly larger role in the predictions compared to the static features, such as proximity, elevation, and fractional water cover. Interestingly, the GRP had relatively low importance for the snowoff model, which is likely due to its erratic behavior during no snow conditions and in areas with significant vegetation cover.

### 5.1.2 Landcover and Uncertainty

Mean absolute errors were binned by landcover features to assess whether these characteristics had a significant influence on model performance. These MAE values were derived from the bootstrapped model predictions, aggregated across all iterations and grouped by landcover type. Landcover features such as elevation, topographic variability, FTC, and proximity, were grouped into four natural breaks and compared with the corresponding model MAE to identify potential patterns or relationships. FW was grouped into three bins including 'low (FW<5%)', 'medium (5%<=FW<10%', and high (FW>=10%). Figure 3 indicates that when elevation, proximity and FW decrease, MAE also decreases for both snowmelt onset and snowoff predictions. Conversely, as FTC increases, MAE also increases, though this is only observable for snowmelt onset predictions. Also notable is that higher FW is associated with higher MAE values for snowmelt onset. FW values exceeding 10% are often indicative of the presence of standing water, riparian zones, or proximity to lakes and rivers (Du et al., 2016). These hydrologically active areas introduce substantial variability in surface emissivity, which can obscure or distort the PMW signal used to detect snow state transitions. Hence, FW may be a major factor behind the overall lower model performance, relative to snowoff. Yet, overall, these landcover and error interactions are as anticipated—error increases with higher surface water cover, coastal proximity, and tree cover.

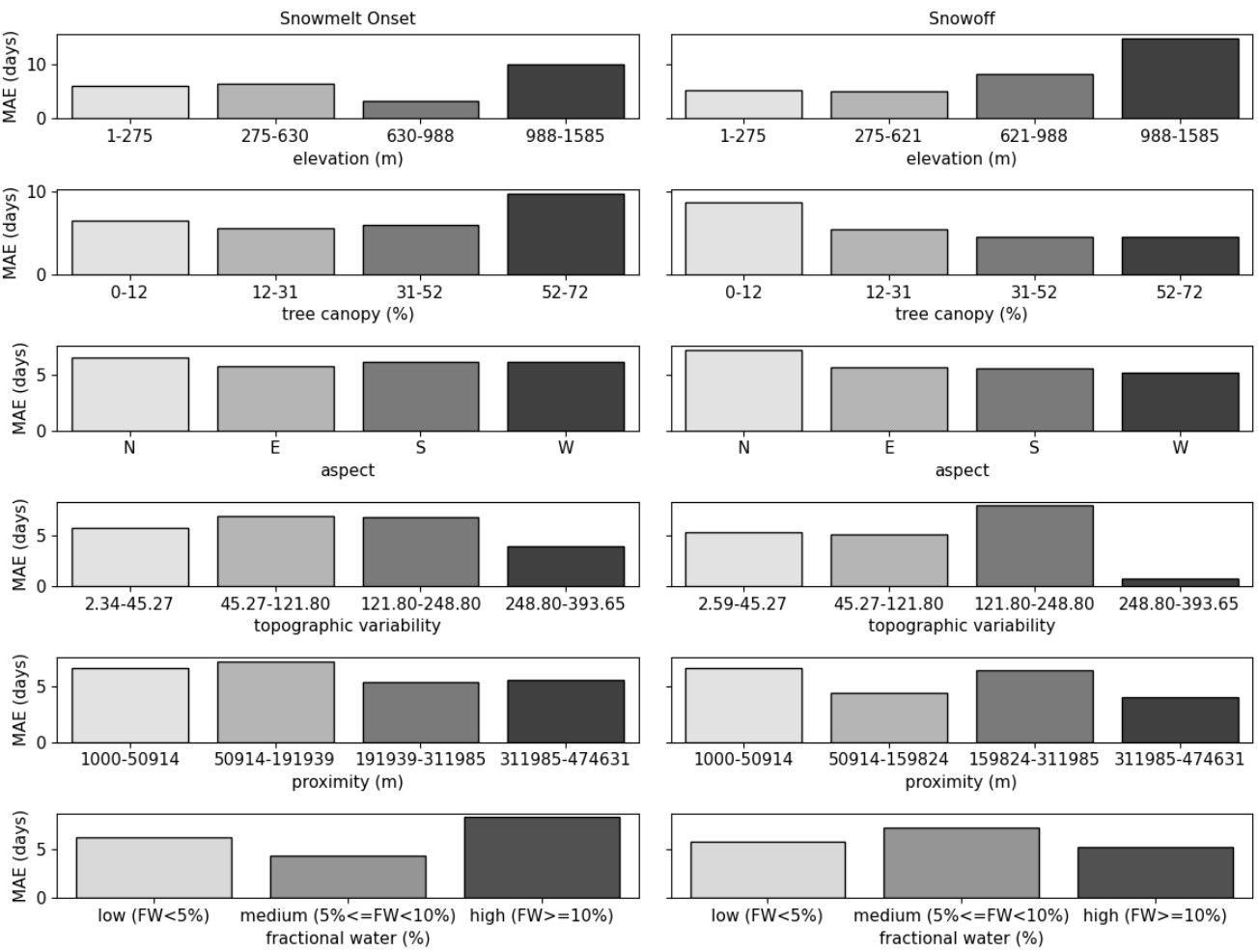

**Figure 3: Average snowmelt onset (left) and snowoff (right) MAE aggregated by landcover features. Landcover was binned using a Jenks classification, except for aspect and FW with lower values on the left and higher values moving to the right.**

The one-way ANOVA results indicate that each landcover characteristic has a statistically significant influence on model error (MAE), with p-values well below 0.05. For both snowmelt onset and snowoff, FW had the largest effect size, with F-statistics of 28.34 and 26.27, respectively. FTC and proximity to water bodies also exerted strong influences on MAE. Specifically, FTC yielded F-statistics of 75.88 for snowmelt onset and 22.62 for snowoff, while proximity showed F = 18.65 for snowmelt

onset and 19.57 for snowoff. Topographic variability and latitude also contributed significantly, suggesting terrain complexity and regional gradients play important roles in prediction uncertainty.

These results indicate that model performance varies significantly with variations in certain landcover features, suggesting that these land characteristics are associated with higher or lower prediction errors. However, ANOVA alone does not demonstrate whether these static land features improve model performance when included as predictors. To test this, we compared RF model performance with and without the inclusion of static variables. When all predictors (both dynamic and static) were used, the $R^2$ values were 0.72 for snowmelt onset and 0.80 for snowoff. In contrast, when static variables were excluded, $R^2$ dropped to 0.64 and 0.40, respectively. These results support the inclusion of static variables as additional RF predictors, despite their relatively low individual importance, as they contribute to overall model performance gain.

## 5.2 Model Comparisons

### 5.2.1 RF and Threshold Comparison

A comparison between the annual median snowmelt onset and snowoff dates derived from the RF model and another established snow phenology record (Pan et al. 2021) is presented in Figure 5. For snowmelt onset, the results show a moderate correlation between the two records for the YRB, with an r-value of 0.58 ($p<0.05$). However, the previous record was derived using a Tb thresholding method and consistently predicted earlier snowmelt onset dates, averaging about 1.5 days earlier than our RF model. When compared to the in-situ testing dataset within the YRB, the previous snow record produced a MAE of 11 days and a RMSE of 14.57 days, similar to our RF model performance.

For snowoff, the two records displayed a stronger correlation, with an r-value of 0.80 ($p<0.05$). The previous approach still showed earlier snowoff bias, averaging about 2 days earlier than the RF-derived snowoff date. Despite the stronger correlation, the thresholding approach returned a high MAE of 32 days and an RMSE of 54 days, which is about double the error derived for the RF method. The RF-derived snowmelt onset and snowoff results also exhibit significantly lower

standard deviations compared to the previous approach, indicating that the ML method is less susceptible to outliers.

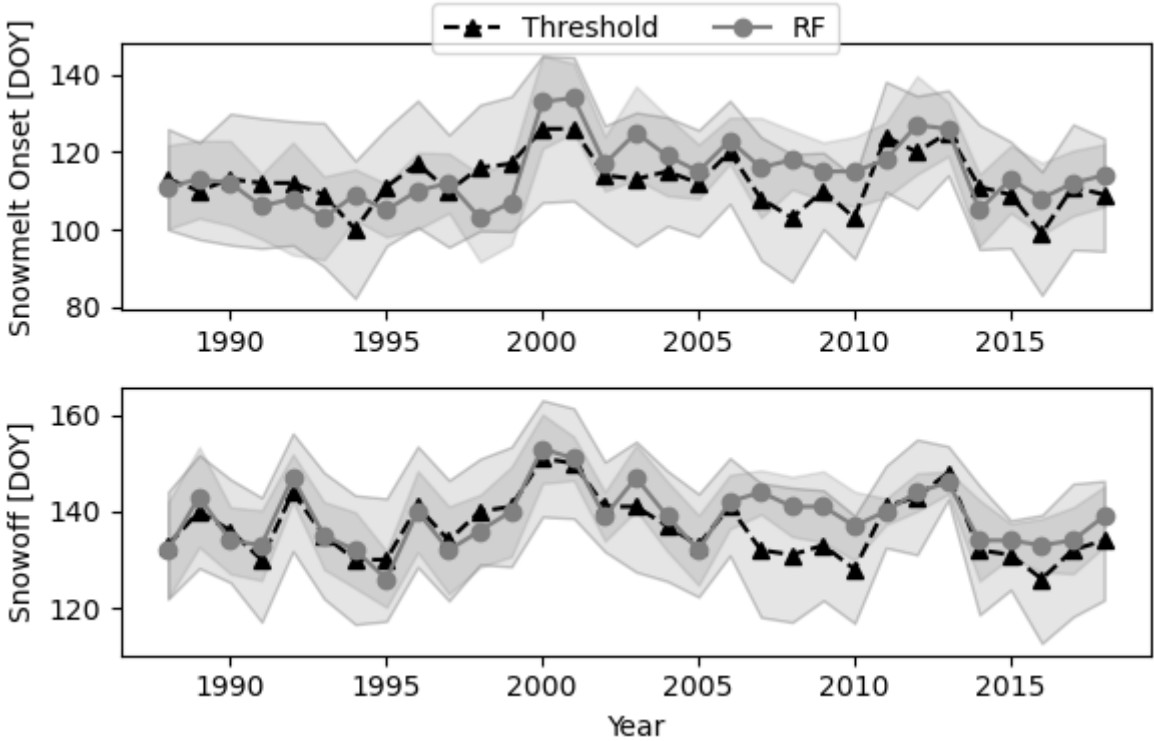

 **Figure 4: 1988-2018 annual median dates +/- one standard deviation for snowmelt onset (top) and snowoff (bottom) for the ABoVE GRP threshold model (Pan et al. 2020) (black triangles) and the RF model (this study) (grey circles).**

### 5.2.2 Snowoff Model Comparison

Our snowoff predictions incorporate two different modeled snow cover datasets, IMS and SnowModel, because no dataset alone spans our full period of record; both datasets were used as features in the RF model. Specifically, we used the 4 km IMS

dataset (2004-2023) and the 3 km SnowModel (1988-2003) to calculate annual snowoff using a 10-day moving window. We then evaluated these outputs against the YRB training dataset to assess their performance relative to the RF results. For the period from 2004-2023, the IMS dataset achieved a MAE of 15.87 days and a RMSE of 21 days. During this same period, the RF-based snowoff dataset achieved a MAE of 16.03 days and an RMSE of 19.2 days.

For the earlier period (1988-2003), the SnowModel dataset produced a MAE of 13.6 days and an RMSE of 16.42 days for snowoff timing. In comparison, the RF snowoff dataset during this period produced a MAE of 18.56 days and an RMSE of 21.2 days. These results reflect the performance of each dataset over their respective timeframes and provide insight into the reliability of the RF-based snowoff model across different periods.

To assess the consistency and variability between the two input datasets, we compared overlapping IMS and SnowModel
derived snowoff dates for three representative years (2008, 2012, and 2016) (Figure A3). These years span a range of snow
conditions during the period of dataset overlap (2004-2020). We found strong agreement in the spatial patterns and
distributions of snowoff timing. For example, the median snowoff date in 2008 was DOY 132 for IMS and DOY 135 for
SnowModel. In 2012, both datasets reported a median DOYof 135, and in 2016, IMS had a median of DOY 126 compared to
DOY 125 for SnowModel. This close agreement suggests that although the IMS and SnowModel datasets originate from
different methodologies, they produce comparable snowoff estimates when applied over the same regions and timeframes.
This general consistency indicates that, our use of these datasets as temporally segmented predictors does not introduce
significant bias and the RF model performance is stable across the full record.

### 5.3 QAQC Maps

The QAQC maps provide a discrete qualitative index for assessing model output quality. These maps were generated by
classifying the predicted model error into four categories – 'Best,' 'Good,' 'Moderate,' and 'Low' – using a quantile
classification. This classification divides the full range of predicted error into bins such that each category contains close to
equal numbers of valid pixels. Because the QAQC maps represent relative predicted error across the domain rather than fixed
error thresholds, a quantile classification scheme was used to ensure consistent and interpretable comparisons between
snowmelt and snowoff model performance (Figure 5). The landcover derived QAQC map for snowmelt onset identified the
mouth of the Yukon and lowlands of the YRB as principal regions of lower model quality. This is likely due to the abundant
small lakes and wetlands in this region, and its comparatively lower elevation and closer ocean proximity, delivering periodic
systems, more ephemeral snowmelt events, and LWC to the snowpack.

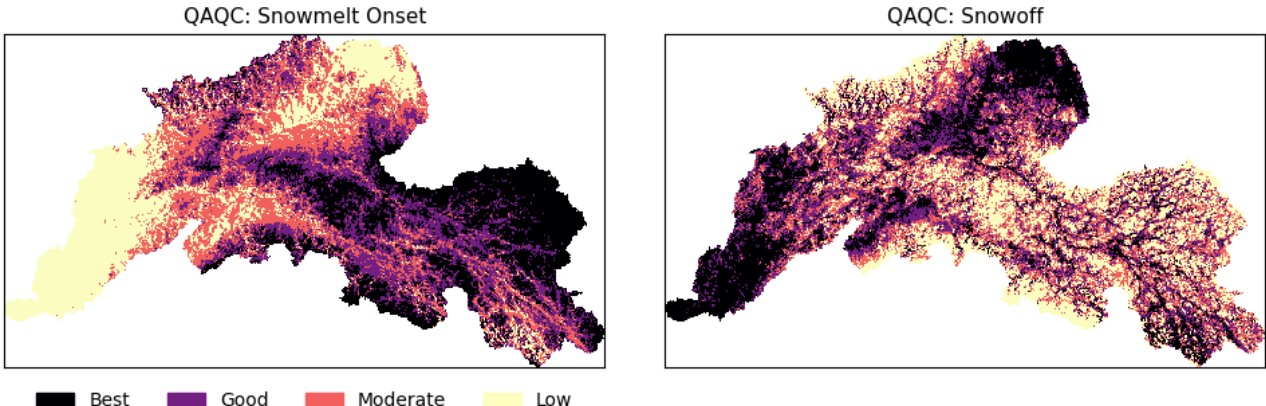

**Figure 5: Snowmelt onset (left) and snowoff (right) QAQC maps were developed to identify regions of relative high to low quality classification results in relation to landcover characteristics.**

In the snowoff QAQC map, the spatial distributions for snowoff differ from snowmelt onset due to differences in the seasonal dynamics of the two events. Snowoff typically occurs more gradually and can span a longer duration, especially in mountainous terrain, where persistent late-season snow and greater terrain and microclimate heterogeneity contribute to larger model error. The snowoff QAQC map identified the upper headwaters of the YRB as having lower quality. However, lower quality pixels are more prevalent at higher elevations and ridgelines. Given that snow cover at higher elevations can linger for extended periods and even through the summer months, it is not surprising that these pixels are ranked as 'low.' The OLS models explained only 35% and 42% of the variability in error for snowmelt onset and snowoff, respectively. The relatively low explanatory power is likely due to the testing data not fully capturing the landscape heterogeneity across the YRB.

## 5.4 Snow Phenology Climatology, Anomalies, and Trends

### 5.4.1 Climatology of Snowmelt Onset, Snowoff, and Snowmelt Duration

The climatology of snow phenology metrics—snowmelt onset, snowoff, and snowmelt duration (SMD)—offers valuable insights into seasonal patterns across the YRB, where snow phenology shows later snowmelt onset and snowoff and longer snow duration in headwaters and higher elevations; this pattern contrasts with generally earlier dates for these metrics at lower elevations and valley bottoms (Figure 6). On average, snowmelt onset (MMOD) occurs around DOY 115 ± 7.7 (~24 April), with the earliest onset recorded on DOY 100 (~9 April) and the latest on DOY 136 (~15 May). Snowoff typically occurs around DOY 139 ± 4.9 (~18 May), with the earliest snowoff observed around DOY 125 (~4 May) and the latest on DOY 168 (~16 June). The snowmelt duration, defined as the period between snowmelt onset and snowoff, spans approximately 22 ± 4.6 days.

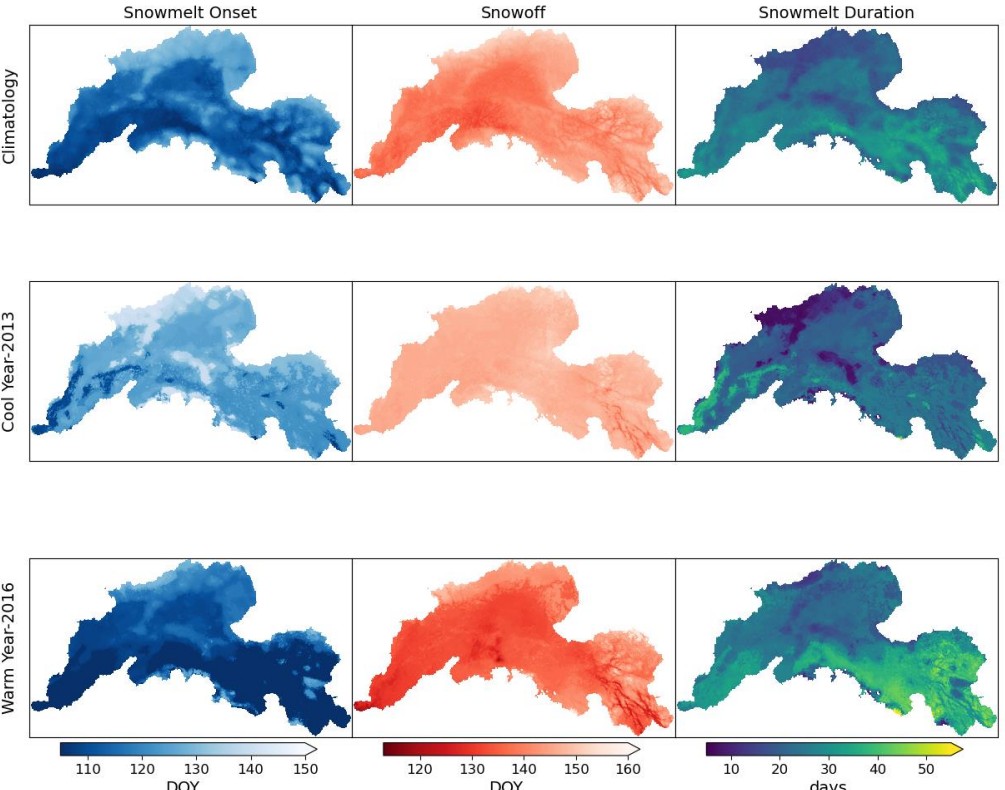

**Figure 6: Climatologies produced for snowmelt onset (left), snowoff (middle) and snowmelt duration (right) for the years 1991-2020 on the top row. Middle and bottom row include the snow phenology for the years 2013 and 2016.**

### 5.4.2 Anomalous Years

Several anomalous years in snow phenology stand out, deviating from the climatological averages. In 2016, a record-breaking warm year (Walsh et al., 2017), snowmelt onset occurred ~9 days earlier than average, and snowoff 6 days earlier, lengthening snowmelt duration by 3 days beyond the mean. Conversely, in 2013, a cooler year, snowmelt onset was 12 days later and snowoff 7 days later, shortening the snowmelt duration by 5 days. These anomalies likely reflect broader climatic drivers, such as temperature fluctuations and abnormal precipitation, affecting snowmelt dynamics in these years. While these deviations are seen in other records (Pan et al. 2021), the RF model successfully reproduced the timing and magnitude of the anomalies, indicating its sensitivity to interannual climate variability.

### 5.4.3 Change in Area Over Time

Temporal changes in area for the 'earlier' class from 1988–2023 (Figure A4) showed no significant correlations for any snow metric. However, the Mann-Kendall Test revealed positive and statistically significant ($p<0.05$) tau values for both snowoff and SMD, though these were modest at 0.2 and 0.25, respectively. To further examine potential trends, we segmented the data into two periods—1988–2005 and 2006–2023—and performed trend analysis on each segment independently (Figure 7).

Annual changes in the snow metric's 'earlier' class during the first half of the data record (1988-2005) identified a strong negative trend toward later snowmelt onset ($r = -0.65$, $p < -0.05$, tau=-0.35, $p < 0.05$). This implies that in the initial period of record, snowmelt onset was occurring earlier across the YRB relative to later years of this period. Conversely, SMD showed a modest positive trend ($r = 0.55$, $p<0.05$, tau = 0.39, $p <0.05$), which suggested a longer snowmelt duration during years with earlier snowmelt onset. Snowoff exhibited no significant trends during this period, with an r value of -0.24, indicating minimal directional change.

In the second half of the data record (2006–2023), annual changes in snowmelt onset displayed a shift to a positive trend, with an r value of 0.54 ($p < 0.05$) and tau trend of 0.42 ($p < 0.05$). This shift suggested that snowmelt onset had been occurring progressively earlier. Snowoff during this period also exhibited a positive trend, with an r value of 0.69 ($p < 0.05$) and tau of 0.50 ($p < 0.05$), indicating an earlier occurrence of snowoff as well. Interestingly, SMD did not show any significant trends during this period due to compensating changes in snowmelt onset and snowoff timing.

To determine whether the temporal patterns we observed were consistent with previous threshold-based snow phenology records, we conducted the same trend analysis using the threshold-based dataset for the period 1988–2018. This record was similarly subdivided into two sub-periods: 1988–2002 and 2003–2018. During the first period, both snowmelt onset and snowoff area exhibited statistically significant negative trends. Snowmelt onset had a Pearson correlation of $r = -0.62$ ($p < 0.05$) and tau $= -0.37$ ($p > 0.05$), while snowoff showed a stronger decline with $r = -0.66$ ($p < 0.05$) and tau $= -0.52$ ($p < 0.05$), indicating a decreasing extent of earlier transitions across the basin. In contrast, SMD during this period showed no significant trend ($r = -0.25$, $p > 0.05$; tau $= -0.18$, $p > 0.05$). For the second period (2003–2018), none of the snow metrics exhibited statistically significant trends. These results suggest that while some temporal changes were evident in earlier decades—particularly for snowoff—the RF model captures more consistent and recent trends (2006–2023).

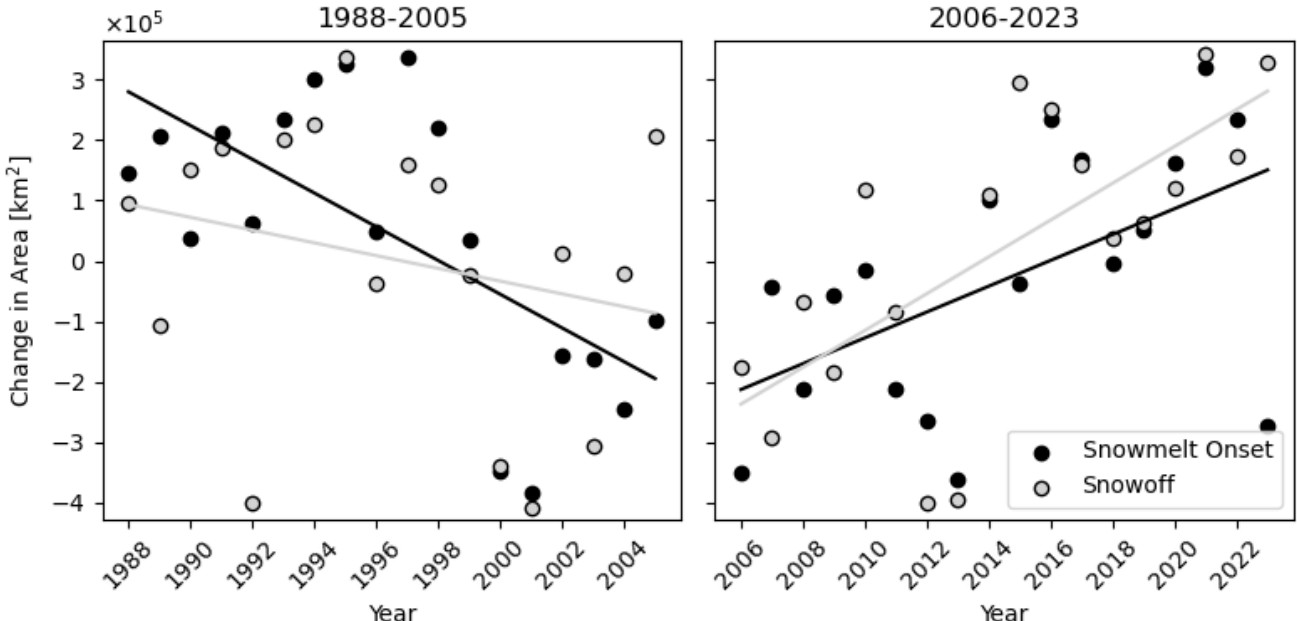

Figure 7: Annual changes in area between the 'earlier' climatology and the 'earlier' class for each year. The left plot shows decreasing trend for snowmelt onset but little change for snowoff from 1988-2005 while the right plot shows significant increase in change in area for both snow metrics from the latter period 2006-2023.

## 5.5 Temperature and Snow Depth Trends Across the YRB

Seasonal temperatures and annual snowfall in the YRB were analysed using in situ measurements from GHCNd stations. For the period 1988–2023, no significant trends were identified in Winter, Spring, Summer, or Spring/Summer temperatures, nor in annual median snow depth on day of snowmelt onset. In the following sections, we present trend analysis results for annual

snowfall and seasonal temperatures across the two sub-periods—1988–2005 and 2006–2023—as well as correlations with the annual snow metrics.

### 5.5.1 Snow Depth at Snowmelt Onset

Between 1998 and 2005, median snow depth on the day of snowmelt onset exhibited a moderately strong negative correlation with time (r = -0.58, p < 0.05; tau = -0.40, p < 0.05), indicating a decrease in snowfall totals during this period (Figure 8). In

contrast, from 2006 to 2023, median snow depth displayed positive and significant correlations and trends (r = 0.68, p < 0.001; tau = 0.40, p < 0.05). We did not examine correlations between snow depth at snowmelt onset and snow phenology metrics, as these analyses would likely introduce bias due to the use of the day of year (DOY) of snowmelt onset in both training and testing datasets.

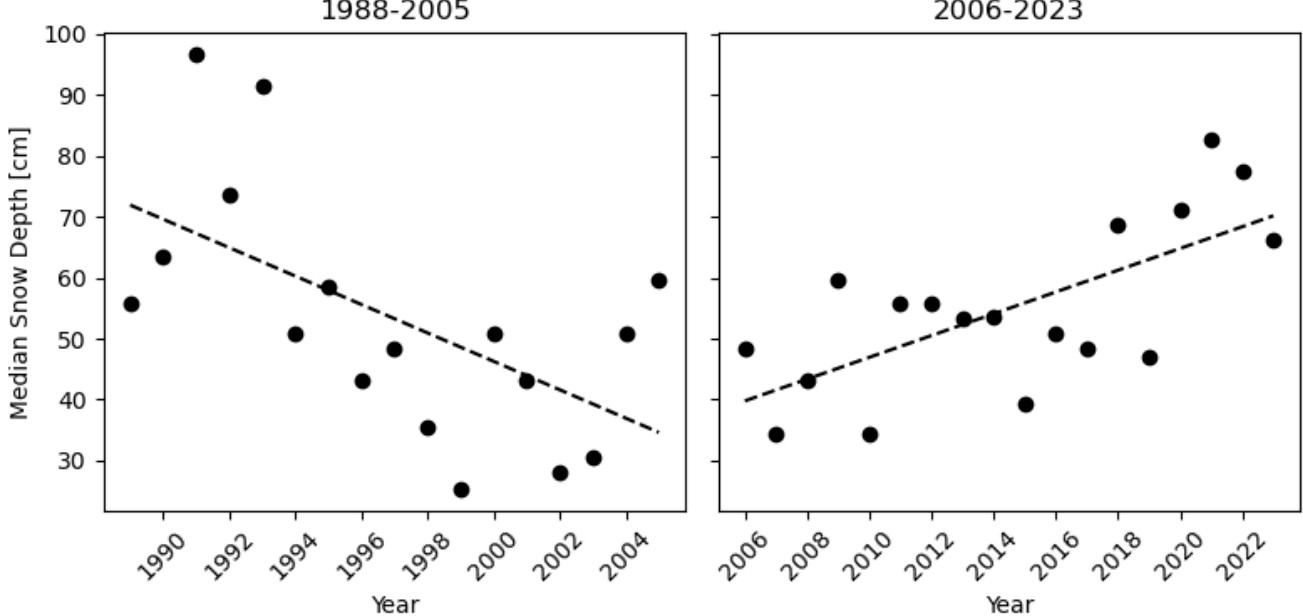

 **Figure 8: Harmonized median snow depth at day of snowmelt onset across the YRB as measured from GHCNd stations.**

### 5.5.2 Seasonal Temperature

Trends in seasonal temperatures from 1988–2005 identified a significant increase in winter temperatures, with an r value of 0.46 ($p < 0.1$) and tau of 0.35 ($p < 0.05$) (Figure 9). Winter temperatures were also negatively correlated with snowmelt onset ($r = -0.47$, $p < 0.05$), indicating that warmer winters were associated with earlier snowmelt. Additionally, snowoff showed a strong positive correlation with spring temperatures ($r = 0.69$, $p < 0.01$). Summer temperatures during this period were moderately correlated with both snowmelt onset and snowoff, with r values of 0.52 and 0.58 ($p < 0.05$), respectively.

From 2006–2023, no seasonal temperatures exhibited significant trends over time. However, winter and spring/summer temperatures had positive tau values of 0.33 and 0.32 ($p < 0.05$), suggesting a slight warming trend. Interestingly, snowmelt onset and snowoff were positively correlated with spring/summer temperatures, with r values of 0.49 and 0.65 ($p < 0.01$), respectively. A significant correlation was also identified between spring temperatures and snowmelt onset ($r = 0.41$, $p < 0.1$), indicating that warmer springs may contribute to earlier snowmelt.

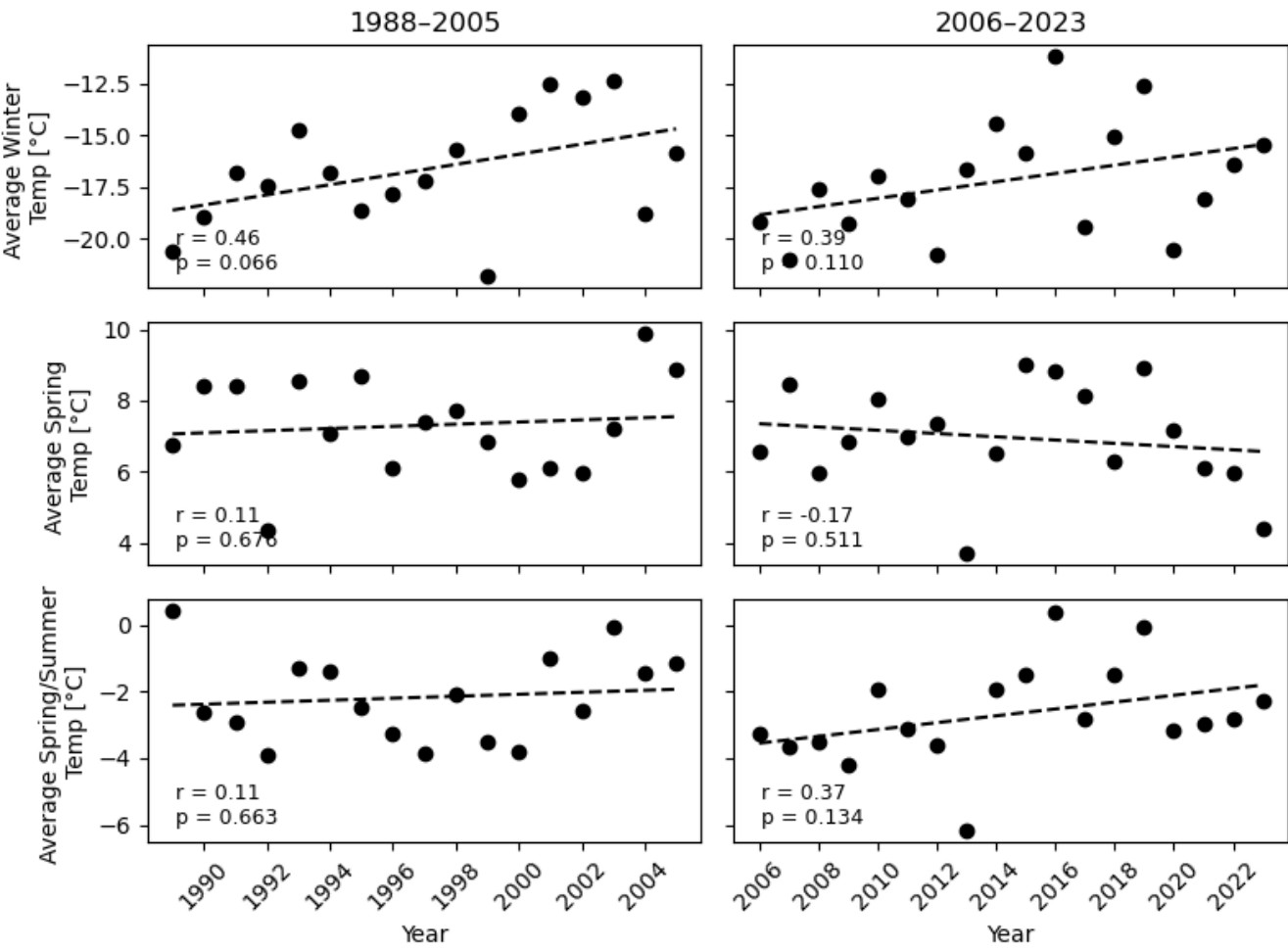

Figure 9: Harmonized seasonal temperatures across the YRB as measured from GHCNd stations.

## 6 Discussion

### 6.1 Model Performance and Limitations

The RF model classified daily snow conditions effectively, achieving high precision and recall scores, underscoring its reliability in predicting snowmelt onset and snowoff. This performance is particularly noteworthy in the complex landscape of the YRB, where traditional threshold-based methods often struggle due to heterogeneous landcover and atmospheric conditions (Pan et al., 2021). By incorporating dynamic timeseries data, such as cumulative TDD and TBD, the model produced favorable predictions of snow phenology events. The inclusion of a temporal dimension within the RF framework further enabled the model to track the seasonal evolution of snow cover, enhancing model accuracy in predicting both snowmelt onset and snowoff.

In comparing bootstrapped performance metrics, the snowoff model outperformed the snowmelt onset model, a result anticipated due to the greater variability and influencing factors associated with detecting snowmelt onset. However, errors calculated by comparing in-situ observations with full model outputs showed that snowoff predictions had a higher MAE and RMSE. Notably, when RF snowoff errors were compared with errors derived from IMS and SnowModel snowoff data, they were found to be similar. This suggests that (1) accurately capturing snowoff at a single point location remains challenging due to high spatial variability at the 3.125 km spatial scale, (2) the RF snowoff model performs on par with other established snowoff datasets and (3) similar uncertainties and bias are also present in other available snow products.

The RF model also faces challenges related to sampling bias, due to the uneven distribution of ground-based snow depth measurements used for training and testing (Tedesco and Jeyaratnam, 2016; Tsai et al., 2019). The GHCNd stations are mostly located in accessible, lower-elevation areas and represent a very small relative local area within a coarser grid cell being evaluated. This also introduces bias into the modeled predictions for underrepresented regions, such as higher elevations or areas of higher FW in the YRB. The scarcity of in situ observations in these areas further limits the ability of the model to generalize across different landcover features. However, emerging technologies, such as camera traps, have shown promise in measuring seasonal snow depth at varying elevations (Breen et al., 2023, 2024). These tools could expand the spatial distribution of snow measurements while reducing potential spatial bias, offering a valuable enhancement for future snow phenology studies.

We found that the RF model predicted snowmelt onset on average 3 days later than an established satellite snow phenology record derived from a Tb threshold method (Pan et al., 2021). This earlier onset predicted by the previous record is likely due to the threshold algorithm misinterpreting seasonal melt events as the main snowmelt onset event. Addressing this misinterpretation was one of the primary motivations for exploring ML in snowmelt onset detection. The RF model, with its logical structure and ancillary data, is likely better able to distinguish between transient melt events and the true seasonal melt onset. Additionally, the coarser resolution of the previous dataset (6.25 km) may introduce bias by failing to capture finer landscape heterogeneity, particularly at higher elevations. These high-altitude areas, which are a smaller portion of the domain, tend to exhibit a lag in spring snow metrics compared to lower elevations. Importantly, the RF model incorporates predictors such as topographic variability and total precipitable water from MERRA-2, which help account for topographic complexity and atmospheric water vapor content—two major sources of Tb error and cloud-related signal contamination in PMW retrievals. This allows for lower ML uncertainty in high-relief regions where terrain-driven heterogeneity and atmospheric interference are common.

The RF models showed greater uncertainty in certain YRB sub-regions, particularly at higher elevations and in coastal areas. These regions likely pose challenges due to their topographic complexity and proximity to large water bodies, which can

introduce both Tb noise and variability in snowpack LWC (Du et al., 2016; Nagler and Rott, 2000). The QAQC maps, highlighted areas with higher RF prediction errors and indicate the need for further model improvementsin these regions.

Incorporating higher-resolution satellite data like SAR and other predictors to better account for landscape heterogeneity and other influential features could improve future iterations of the model and address some of the scale dependent uncertainties (Darychuk et al., 2023; Gagliano et al., 2023; Marin et al., 2020).

Although the RF model achieved MAE values like or slightly higher than those from previous threshold-based methods, it

offers distinct advantages not reflected in summary metrics alone. The bootstrapping results indicate that when the prediction data are well-represented within the training dataset—such as under similar climate or landcover conditions—the RF model yields substantially lower errors. This highlights the model's ability to generalize effectively when sufficient representative training data are available.

Furthermore, the RF approach reduces false detections of snowmelt onset caused by transient warming events by learning from multivariate seasonal context, rather than relying on fixed thresholds. It also performs more reliably in heterogeneous and hydrologically complex regions, such as areas near lakes or coastlines, where PMW land signals are often confounded by water contamination. Finally, the RF flexibility allows for seamless integration of new predictors, such as SAR data or reanalysis-based forcings, enabling continuous model refinement. Together, these strengths—daily classification of snow state,

spatially explicit uncertainty, improved seasonal tracking, and enhanced adaptability—represent a meaningful advancement over traditional thresholding methods for operational snow monitoring in the YRB.

**6.2 Implications of Changes in Snow Phenology**

Annual changes in snow phenology are closely tied to current climate conditions, with snowmelt onset and snowoff generally occurring later in cooler years and earlier in warmer years. Notably, during warm years, snowmelt occurred earlier—by ~9

675 days—while in cooler years, the delay was ~12 days, relative to the climatology. And the difference in snowmelt onset between a warm year and cool year could be as much as 21 days. Earlier occurrence in snowmelt onset during warm years also generally translated into a longer melt duration, as noted from previous studies (Musselman et al., 2017).

Our analysis revealed that snowmelt onset trended toward a later date from 1988–2005, a result that might seem

counterintuitive given the warming temperatures typically observed at northern latitudes. However, during this period, annual snowfall was particularly high in the beginning of this period (especially before 1994), and seasonal temperatures showed small changes. The deeper snowpack meant that more snow was available for melting earlier in the season, as reflected in the RF model outputs. This suggests that snowmelt onset is influenced more by the quantity of accumulated snow than by

temperature alone (Barnett et al., 2005; Frei and Henry, 2022; Trujillo et al., 2012), except in cases of anomalously warm years, where elevated temperatures tend to override other factors (Musselman et al., 2017).

With temperatures remaining relatively stable from 1988-2005, snowoff timing followed typical seasonal patterns, explaining why we observed no significant changes during this period. Here we demonstrated the important role winter snowfall in shaping springtime snow phenology, yet trends in snowfall patterns across Alaska are complex. From 1957 to 2021, winter snowfall equivalent has increased across Alaska. However, northern and southern regions have seen a decline in snowfall during the spring and fall shoulder seasons, effectively shortening the snow cover duration (Ballinger et al., 2023).

We also showed an acceleration toward earlier snowmelt onset and snowoff timing during the latter half of the data record (2005-2023). These changes align with recent warming trends in average Spring/Summer air temperature measurements in the YRB. Over the last century, Alaska has experienced varying increases in temperature across different climate divisions (Bieniek et al., 2014), with record-breaking warmth in recent years (Lara et al., 2021; Swanson et al., 2021; Walsh et al., 2017).

## 7 Conclusion

This study introduced an application of a RF approach to derive a 35-year snow phenology record across the YRB, delivering new insights into the timing and variability of seasonal snowmelt onset and snowoff across Alaska's largest drainage basin. Designed for enhanced delineation of spatial and temporal heterogeneity in snow metrics over more established satellite and model data records, the RF model effectively classified daily snow conditions, achieving reasonable accuracy in delineating snow phenology metrics across a highly varied landscape. By working with an improved spatial resolution of 3.125 km, the RF model was able to provide a more detailed representation of landscape features than previous Tb threshold-based snow phenology datasets, supporting more precise predictions across the YRB's diverse terrain. The enhanced spatial resolution proved especially valuable in depicting snow phenology in the region's remote, high-latitude environments, allowing the RF model to capture nuances in snowmelt timing across varying topographies, elevation ranges, and vegetation covers.

One significant advantage of the RF approach over traditional thresholding methods is the reduction in sensitivity to transient melt events and atmospheric fluctuations, making a more reliable identification in primary snowmelt onset rather than temporary thaw and early melt events caused by brief warming episodes. Additionally, the integration of dynamic predictions within the RF model, including cumulative thaw degree days and snow cover presence, allowing the capturing of the seasonal evolution of snow conditions, while remaining less prone to errors related to isolated atmospheric warming events. However, certain challenges emerged, particularly in areas with complex topography, such as in high-elevation and coastal zones, where model prediction errors were greater. These challenges likely reflect the difficulties in addressing highly localized factors, like changes in snowpack liquid water content, or terrain induced microclimate variability.

As with many remote sensing models, sample bias in the RF model due to uneven ground-based data coverage poses a limitation, as in-situ snow depth measurements are predominantly collected in accessible, lower-elevation regions. This bias suggests the need for continuous updates to the in-situ training dataset, particularly by expanding measurements in higher-altitude and coastal areas within the YRB. Incorporating more extensive in-situ observations would improve the model's accuracy in underrepresented regions, allowing for a more comprehensive understanding of snow phenology across the YRB. By overcoming these current limitations and incorporating higher-resolution remotely sensed data sources, such as SAR, future iterations of the model could further enhance snow phenology monitoring in the YRB, making it a critical tool for understanding snow-related dynamics in response to climate change.

Finally, this study produced an extended snow phenology record spanning more than 30-years to better distinguish climate normals and quantify long-term climate trends in the YRB. By segmenting the 35-year record into two timeframes (1988–2005 and 2006–2023), we were able to detect distinct temporal trends in the spring snow metrics that corresponded with changes in seasonal temperatures and snowfall patterns. The analysis revealed that in earlier years, snowmelt onset tended toward later dates, influenced largely by higher snowfall amounts and stable seasonal temperatures. However, in more recent years (2006–2023), both snowmelt onset and snowoff timing have advanced significantly, coinciding with rising spring and summer temperatures across the YRB. These phenological shifts, along with the lengthening of the snow-free season, align with observed patterns of earlier spring onset and more frequent anomalous warming events in recent years. The resulting snow phenology trends offer valuable insight into the YRB's changing climate and highlight the increasing influence of warming on snowpack dynamics, which hold implications for regional water availability, ecosystem health, and community resilience in the face of accelerated climate change.

**8 Appendix A**

**Table A1. Descriptions of datasets used in this study and their sources.**

| Dataset | Spatial Resolution | Spatial Domain | Temporal Resolution | Period of Record | Use | Reference/Source |
|---|---|---|---|---|---|---|
| 19 V and 19 H (K-band) | 6.25 km | Northern Hemisphere | Daily | 1988-Present | RF Prediction | Brodzick et al. 2018 |
| 37 V and 37H (Ka-band) | 3.125 km | Northern Hemisphere | Daily | 1988-Present | RF Prediction | Brodzick et al. 2018 |
| Daymet | 1 km | OCONUS | Daily | 1980-Present | RF Prediction | Thornton et al. 1997 |

| | | | | | | |
|---|---|---|---|---|---|---|
| IMS | 4 km | Northern Hemisphere | Daily | 2004-Present | RF Prediction | Helfrich et al. 2007 |
| SnowMod | 3 km | Alaska and NW Canada | Daily | 1980-2020 | RF Prediction | Liston et al. 2023 |
| MERRA2 TQV | 50 km | Global | Daily | 1980-Present | RF Prediction | Gelaro et al. 2017 |
| Fractional Water (FW) | 6.25 km | Alaska | Static | 2003-2015 | RF Prediction and Uncertainty Analysis | Du et al. 2017 |
| Fractional Tree Cover (TC) | 250 m | Alaska | Static | 2011 | RF Prediction and Uncertainy Analysis | Carroll et al. 2011 |
| Elevation (GTOPO) | 1 km | Alaska | Static | | RF Prediction and Uncertainty Analysis | USGS |
| Elevation (ALOS) | 30 m | Global | Static | 2006-2011 | RF Prediction and Uncertainty Analysis | Tadona et al. 2014 |
| Proximity | 1 km | Alaska | Static | | RF Prediction and Uncertainty Analysis | GTOPO |
| Aspect | 1 km | Alaska | Static | | RF Prediction and Uncertainy Analysis | GTOPO |
| Glaciers | vector | Alaska/Canada | Static | | Indicate permanent ice | Pfeffer et al. 2014 |
| GHCNd | in situ | Alaska | Daily | <1988-Present | RF Prediction - Testing/Training | Menne et al. 2012 |
| MMOD | 6.25 km | Alaska | Annual | 1988-2018 | RF Comparison | Pan et al. 2021 |
| Snowoff | 6.25 km | Alaska | Annual | 1988-2018 | RF Comparison | Pan et al. 2021 |

**Table A2. Gridsearch results for RF hyper parameters.**

| | Values Tested | Snowmelt Onset | Snowoff |
|---|---|---|---|
| n_estimators | 8, 100, 200, 300, 1000 | 100 | 100 |
| min_samples_leaf | 1, 2, 4 | 4 | 4 |
| max_depth | None, 10, 20, 30 | 10 | 10 |
| max_features | auto, sqrt, log | log2 | log2 |
| min_samples_split | 2, 5, 10 | 5 | 10 |

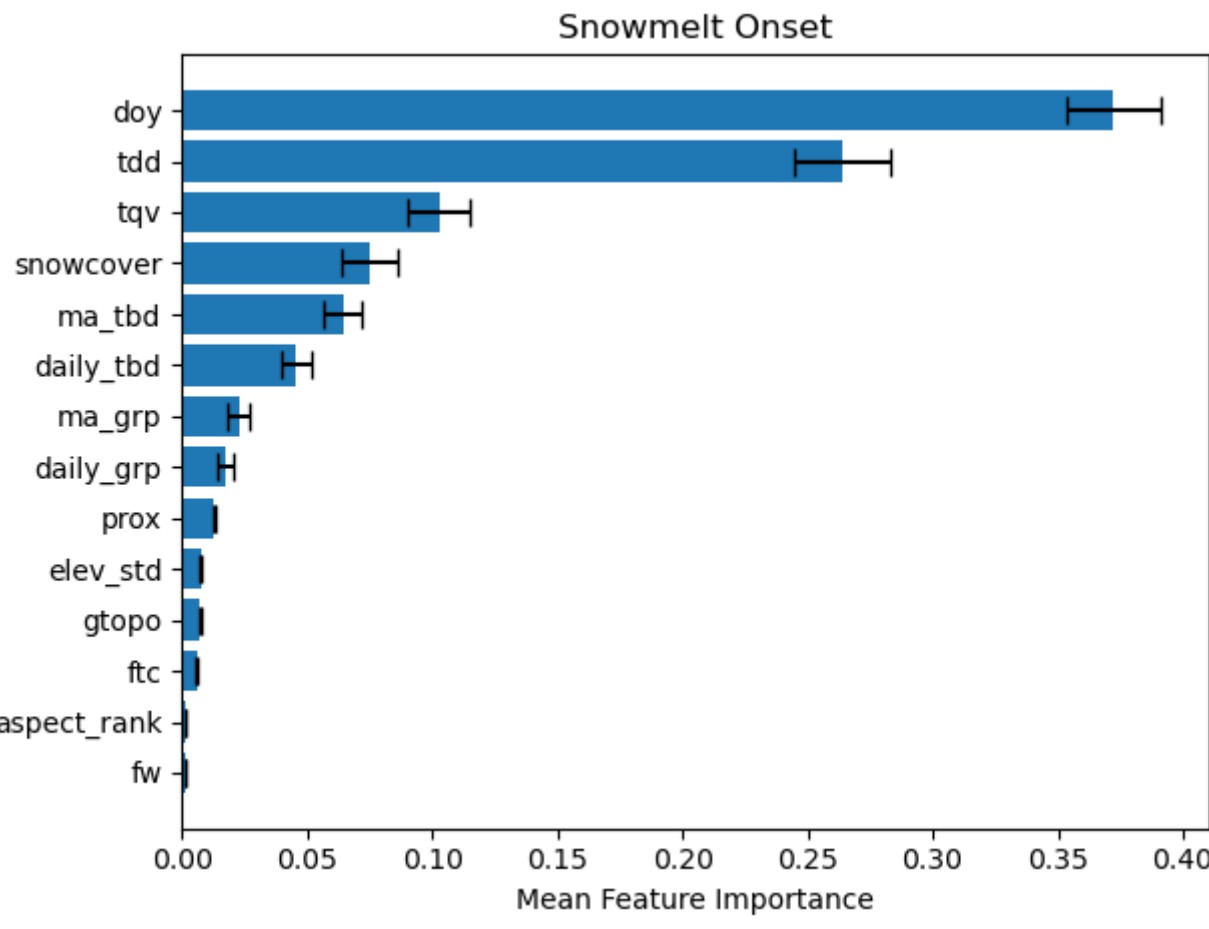


**Figure A1. Snowmelt onset variable feature importance with +/- one standard deviation over 20 bootstrap iterations.**

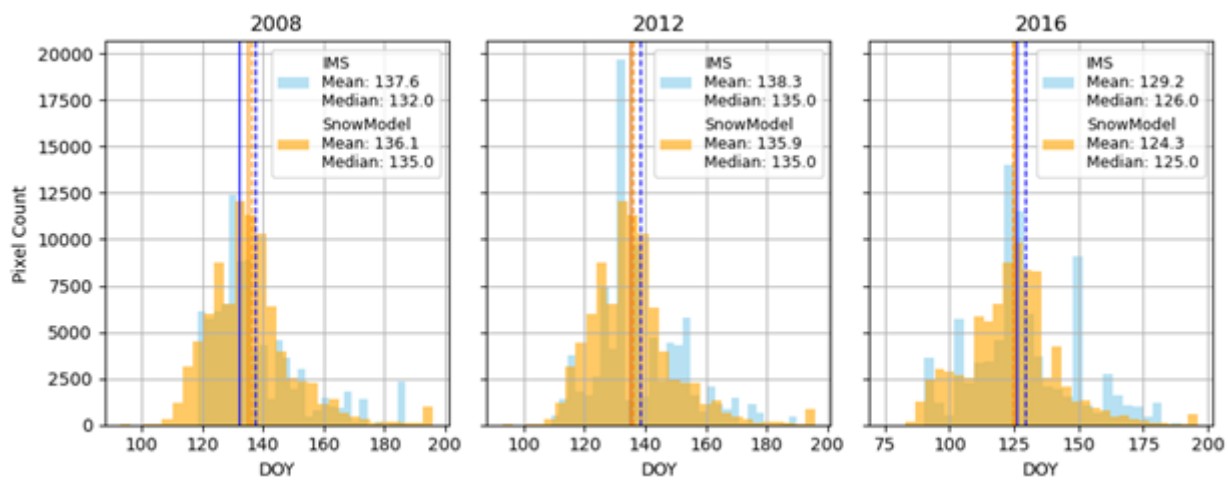

**Figure A2. Snowoff variable feature importance with +/- one standard deviation over 20 bootstrap iterations.**

**Figure A3. Histogram comparison of snowoff detection from SnowModel (orange) and IMS (blue) over three overlapping years including 2008, 2012, and 2016.**

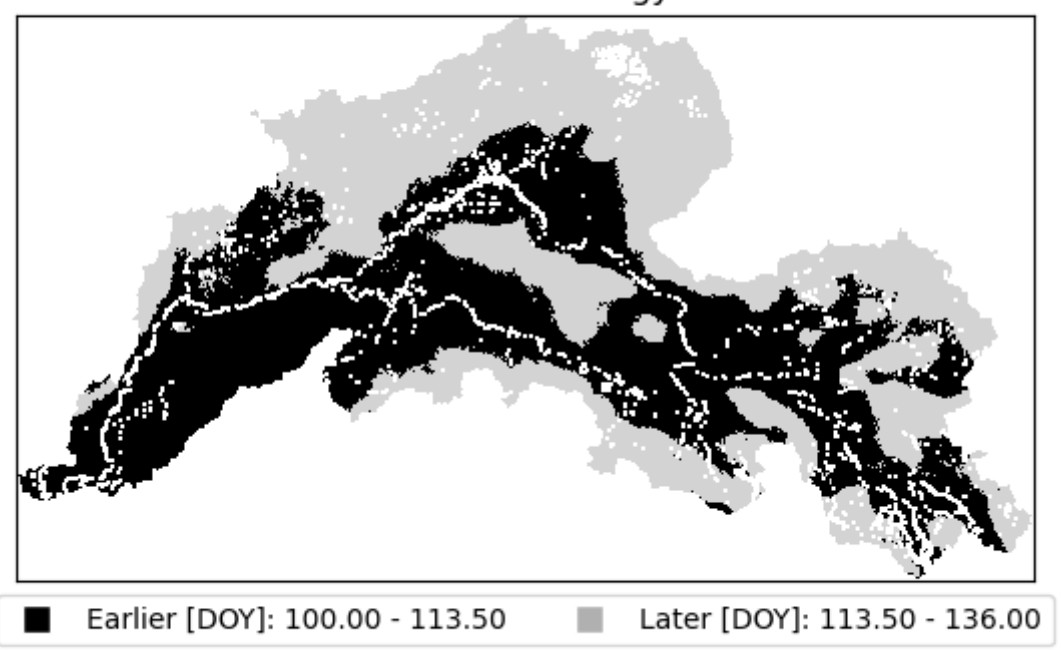

Earlier [DOY]: 100.00 - 113.50   Later [DOY]: 113.50 - 136.00


**Figure A4. Snowmelt onset climatology binned into two classes, 'earlier' and 'later.' This climatology was used to assess annual changes in snowmelt onset.**

*Code and Data availability.* Code and datasets produced in this study are available upon request.


*Author Contributions.* CGP wrote the manuscript. CGP processed data and analyzed the results. CGP, KL, SPG, JSK, JD, PBK contributed to the design and conceptualization. All coauthors contributed to writing and editing of the manuscript. All coauthors have read and agreed to the published version of this manuscript.

*Competing interests.* We declare no competing interests are present.

*Acknowledgements.* We thank Benoit Montpetit and Devon Dunmire for their constructive reviews and contributions, which helped improve this paper. This research was funded by the U.S. Army Corps of Engineers, Engineer Research and

Development Center (ERDC) under PE 0602146A/AT9, Project 'Tactical Geospatial Information Capabilities', Task
'Geospatial Analytics and Prediction'. Permission to publish was granted by the ERDC Public Affairs Office. Any opinions
expressed in this paper are those of the authors and are not to be construed as official positions of the funding agency.

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
