# Peer review of "A random forest derived 35-year snow phenology record reveals climate trends in the Yukon River Basin"

_EGUsphere, 2024_

## Referee Comment (RC2)

**A random forest derived 35-year snow phenology record reveals climate trends in the Yukon River Basin**
Pan et al (2025)

This manuscript uses a random forest to estimate snowmelt and snowoff timing in the Yukon River Basin, Alaska. First, I apologize to the authors for my delay in submitting these comments. The manuscript is generally well written, but would benefit from additional details provided in the methodology. In general, I am not convinced that this new methodology is a substantial improvement compared to already existing methods. As such, I believe that major revisions are necessary before publication.

**Major comments**

My major concern with this work is that it seems that the RF approach isn't really necessary. In fact, for snowmelt onset, the previous thresholding approach appears to do better (MAE of 11 days compared with 11.6 days for RF), and the IMS dataset does better for snowoff timing (MAE of 15.87 days compared to 16.03 days for random forest). Why can a combination of the thresholding approach for snowmelt onset, and IMS for snowoff not be used then?

Further, In the conclusion, it is stated that this study delivers "new insights into the timing and variability of snow melt onset and snowoff" but I actually fail to see these new insights. A lot of this phenology was already presented in Pan et al 2020 (i.e. section 3.2). Is there actually any new insights that are gained from the RF approach, that are absent when using the thresholding method? The motivation for this work needs to be substantially improved. Given the previous work from 2020, which already presents a 30-year snow phenology record of the YRB with many of the same insights, what does this work add? Why is current knowledge of snow-melt and snow-off conditions in YRB insufficient?

Second, some details are lacking about the data/methodology. Also, in general, a schematic of the workflow would be incredibly helpful. Please see the below comments, organized by section:

Section 3.2.1
L165 – How frequent are missing temporal observations in this data?

Section 3.2.2
Isn't there also a 1km IMS product that exists? Why was this not used? I also find details about SnowModel to be lacking here. What is the model forced with? Is it consistent with IMS when the two products overlap?

Section 3.2.3
How was the cumulative thaw degree days calculated? Cumulative from what point in time?

Section 4.1

Did you try any other ML models such as xgboost or neural networks?

Section 4.1.1
How many combinations of parameters did you try in the grid search? What were the parameters that were changed and what was the range of values tested? Please also reference Table A2 in this section.

Section 4.1.2
More details in this section would be helpful. For example, how was the split between training and testing done? How was the feature importance computed? Why was the QC metric split into 4 categoric categories, instead of just kept as a number between 0 and 1? Is there information that is lost in doing this?

Section 4.2.1
The classification into 'earlier' and 'later' snow events is not clear. What does 'snow event' mean specifically? Why was this done?

Section 4.2.2
Please specify what this data was used for.

**Minor/technical comments:**

L 66: "While higher-frequency K-band…" Perhaps this line should go to the next paragraph? It seems a bit out of place in this paragraph which primarily covers C-band SAR and Sentinel-1.

Figure 1 – Please change the triangle markers to dots.

Section 3.1 - This study is for the YRB, but many of the sites used are located outside the YRB, particularly a large number along the southern coast. I understand this is likely done to enhance the limited in-situ data available to train the model, but the impact of this this should be mentioned somewhere.

L140 – What is meant by 'analysts'?

L144 – Why is there a different number of observations available for training snow melt onset and snowoff?

L149 – please specify that the snow cover is binary.

L146 – 155 – Please mention somewhere in these paragraphs that these variables will be described in more detail in the following sections.

L153 – "GTOPO" in parenthesis is not needed.

L172 – 174 – please change "reduce" to "reduced" in these sections to be consistent with the tense throughout the methods section. Check other areas in the methods as well.

L176, L178 – The equations are numbered in the text, but not in-line. If possible, please include a number next to the equations for easy reference.

L209 – Can the 'TC' abbreviation be changed to 'FTC' to be consistent with 'FW' ("fractional water")?

L226 – I do not fully understand the sentence: "Although RF does not inherently model temporal sequences like some other algorithms, temporal dimensions were incorporated by structuring each day as a sample within a sequential framework." Specifically, the 'within a sequential framework' is throwing me off. RFs don't have a sequential framework, and each sample is treated distinctly, not as part of a sequence. Can you please clarify this here?

L230 – what were the spatial and temporal resolution of the other similar snow record? Is it really correct to say that this study is a 'significant temporal enhancement'?

L239 – "By labeling the timeseries accordingly…" It is not clear to me how labeling the timeseries this way enables any of the following numbered points.

Figure 2 – In the bottom panel, it would be more clear to set the y-axis as merely "dry" for 0, and "wet" for 1.

L302/306 – Are these the scores of the training dataset? testing?

L315 – Is there any bias in the snowmelt/snowoff timing?

L320 – What is "the full YRB dataset"?

L322-323 – It is not clear to my how these MAE values differ from the ones presented in L315-317.

Section 5.1.2 – What are you using as the ground truth in this section to compute the MAE?

L335 – the 'FW' feature was not grouped into 4 natural breaks (not a variable in Fig. 3)

L339 – "higher FW is associated with much higher MAE values for snowmelt onset." I don't see this in Figure 4. FW of 10 and 13% have by far the highest MAE and then it decreases for higher values. Maybe I'm misunderstanding this figure?

Figure 3 – I think the division into 'four natural breaks' is not super helpful in this figure. I guess we can see that MAE for snowmelt is higher at higher elevation but what are the

elevations? Also, it doesn't make sense for the aspect feature as aspect describes a direction that has a cyclical nature, not really a standard increasing variable.

Figure 4 – The x-axis for this plot is confusing to me. What is it binned so seemingly randomly? For example, the x-axis jumps from 10 to 11 to 13 to 22.

L341 – Why is it anticipated that error increases with higher surface water cover?

L352 – "These results support inclusion of the static variables as additional RF predictors, despite their relatively low importance". I'm not totally convinced by this. All the ANOVA results are saying is that the MAE is statistically significantly different for different land cover types, not that the land cover type is helping improve the model MAE, right? This analysis could indicate that perhaps your model is missing processes for certain land cover types, thus influencing the MAE for these land cover types, but I don't think it indicates that the static variables are helping improve the model. Did you test models with these static variables omitted to see how the MAE changes? This would give a better indication that the static variables are useful.

Section 5.4.2 – Are these anomalies better captured by the RF or are they already seen in other records?

L421 – "snowmelt onset occurred ~9 days earlier than average, and snowoff 6 days earlier, lengthening snowmelt duration by 4 days…" Maybe this is a rounding error, but should this not be 3 days? Same with the following sentence. Maybe including decimals would help.

Section 5.5.2 – Can a figure be included for the trends in seasonal temperature? Also, what does the 'tau' value represent?

A figure comparing the performace of the different methods (IMS, SnowModel, thresholding, RF) for snowmelt and snowoff would be helpful.

---

## Author Response (AR1)

Response to Reviewer 1:

We thank Dr. Montpetit for taking the time to read our manuscript and provide a constructive and thorough review. We did our best to address each comment and believe the revised manuscript will be greatly improved. Our responses to each comment were highlighted in pink.

**1. General Comments**

This paper calibrates a Random Forest (RF) model using dynamic and static variables to identify snow phenology. Among the dynamic variables, the authors use the MEaSUREs passive microwave (PMW) brightness temperatures (TB) and modeled snow cover (IMS/SnowModel). They then apply the RF classifications over a 35-year period to identify phenology trends.

The methodology is sound, and the manuscript is generally well written. However, some clarifications are needed to improve the manuscript, and a few details still require discussion and analysis. That said, this study provides valuable insights and will benefit the cryosphere research community.

Thank you for the overall positive review.

Although RF is an effective tool for multi-variate predictions, significant work has also been done using Neural Networks (NN), which are now widely accessible and have proven to be more robust in linking snow properties to PMW measurements (Forman and Xue, 2017). While it is not necessary to reprocess this study, NN should be considered for future implementations. NN are particularly effective at simultaneously identifying spatial and temporal relationships within datasets, which could benefit this study.

We agree with your comments. We will discuss the use of alternative machine-learning methods in particular NN in the revised manuscript. DONE

That said, Xiao et al. (2021) did show that RF outperformed Artificial Neural Networks (ANN) in retrieving fractional snow cover (FSC) using the MEaSUREs TB dataset. This latter reference is relevant and should be cited in the manuscript.

We will cite Xiao et al. 2021 to our text on RF application in the cryosphere. DONE

Finally, some questions remain regarding the accuracy of the calibrated MEaSUREs TB dataset. Since this study does not use radiative transfer modeling, the TB dataset is still appropriate; however, the calibration uncertainty should be addressed within the RF model and compensated for by including additional input variables.

**2 Major Comments**

The first comment I have is in the processing of TBs. Most studies using TBs at those frequencies will apply an atmospheric correction to remove the impact of the atmospheric water vapour. Dolant et al. (2016) mentions that at 37 GHz, the atmospheric contribution to the measured TB can be up to 10 K. This is 20 ~ 10 % of the signal which is not to be neglected. It will also have a significant impact on the RF model predictions since no atmospheric information is included in the predictor variables. I strongly suggest correcting TBs with the methodology of Liebe (1989), or at the very least include atmospheric

water vapour from a reanalysis dataset as an input variable to the model to account for atmospheric conditions.

In the updated manuscript, we will incorporate total column water vapor from the MERRA-2 reanalysis product as an additional dynamic predictor variable in the RF model. This variable will be extracted at 6 PM Alaska Time to align temporally with the evening overpass of the satellite sensors used to derive TBs. Including this predictor allows the RF model to account for atmospheric variability without explicitly correcting the TBs, preserving consistency with previous studies while improving model robustness.

We will also add a discussion in the methods and limitations sections of the revised manuscript to address this point directly and clarify how atmospheric effects are now considered in our modeling framework.

DONE

Another concern I have about this study is that, it was shown by Meloche et al. (2022), using an RF model, that snow depth is strongly linked to topographic features. In this study, it seems it has little importance. I would suggest including similar topographic features from high-resolution DEMs to ensure that topographic information is well captured by the RF model. There are also known issues with TB measurements in high-topography areas (Xiong et al., 2022), and this could help improve the predictions in this study.

This leads to another concern, using TB data in high-topography areas. TB measurements in those areas show high errors due to many factors, one being local incidence angle variation. As mentioned above, adding more topographic information could reduce the uncertainty and could improve the RF model reliability in these areas.

We thank the reviewer for raising these important and insightful points. We fully agree that topography plays a critical role in snow distribution and microwave signal behavior, particularly in complex mountainous terrain.

In our original implementation, we used the GTOPO30 digital elevation model, which has a spatial resolution of approximately 1 km. We acknowledge that this coarse resolution may have failed to capture key variations in topographic features across the region, thereby limiting the RF model's ability to leverage terrain-related predictors effectively.

In response to your recommendation, we will incorporate higher-resolution topographic data derived from the ALOS World 3D-30m DEM. From this data, we will calculate additional terrain metrics such as the standard deviation of elevation and/or a topographic complexity index aggregated to the 3.125 km EASE-grid pixel scale used in the study. These metrics will provide an indication of sub-grid topographic heterogeneity, which is particularly relevant in mountainous areas where elevation changes rapidly within a single coarse-resolution grid cell.

By enhancing the topographic inputs to the model, we aim to better capture the terrain-driven variability in snow accumulation and TB behavior. This refinement is also expected to mitigate known TB retrieval uncertainties in high-relief regions, such as those caused by variable local incidence angles (Xiong et al. 2022). These additions should improve the reliability and spatial representativeness of the RF model across complex landscapes.

DONE

**3. Minor Comments**

**Line 50:**
It is not accurate to state that passive microwave sensors (PMWs) are insensitive to cloud cover. Please revise this sentence in accordance with the major comment above. PMWs can penetrate cloud cover; however, they are still affected by clouds and the total atmospheric water column. Atmospheric corrections should be applied.

We will modify this sentence to: 'The moderate frequency (~≤37 GHz) retrievals from operational satellite microwave radiometers are sensitive to snow cover conditions, providing nearly continuous, year-round data. Importantly, the propagation of microwave energy through the snowpack is responsive to changes in snow structure, including liquid water content (LWC), grain size and density, which are key indicators of snowmelt onset (Tedesco et al., 2015).' Removing the incorrect description of PMWs being insensitive to cloud cover.

DONE

We will also address the atmospheric impacts in a newly added section 3.2.4 as follows:.

3.2.4 MERRA-2 Reanalysis

Precipitable water vapor and precipitating clouds can affect PMW observations (Du et al., 2014) and assert adverse effects on snow retrievals (Dolant et al. 2016). To mitigate atmospheric influence on SSM/I and SSMIS evening observations, the water vapor estimates for 6 PM Alaska Time were extracted from the MERRA-2 Reanalysis dataset and integrated in the RF framework. The MERRA-2 data were collected from the NASA EarthData Explorer and resampled to 3.125 km using a nearest neighbor interpolation.

Added reference:

Du, J., Kimball, J.S. and Jones, L.A., 2014. Satellite microwave retrieval of total precipitable water vapor and surface air temperature over land from AMSR2. IEEE Transactions on Geoscience and Remote Sensing, 53(5), pp.2520-2531.

DONE

**Lines 66–68:**
This sentence is somewhat confusing. The paragraph discusses active microwave systems, but this sentence feels like it belongs in the previous paragraph.

We agree. We will remove this sentence and place it at the end of the previous paragraph.

DONE

**Line 71:**
Please justify the choice of only using SSM/I and SSMIS in the study. Why were AMSR, AMSR2, and SMMR not included to extend the time series and add more data?

Thank you for this important comment. We chose to use SSM/I and SSMIS because they are both part of the Defense Meteorological Satellite Program (DMSP) with similar sensor configurations, and the NSIDC MEaSUREs dataset provides a consistent, well-calibrated dataset with stable acquisition times and long temporal coverage. This consistency is particularly valuable for long-term trend analysis and ensures compatibility in terms of orbital characteristics, overpass times, and calibration standards.

We excluded SMMR due to significant data gaps and dropouts that we observed during preliminary processing. Incorporating SMMR would have required substantial temporal interpolation and gap-filling, which could introduce additional uncertainty and reduce the robustness of the training dataset. As for AMSR and AMSR2, although these sensors provide valuable data, their records do not fully overlap with the long-term SSM/I–SSMIS time series, and their inclusion would require additional cross-sensor calibration steps to ensure consistency across the record (e.g., the need to account for different spatial resolutions and observation geometry). To maintain continuity and data integrity over the full 35-year analysis period, we focused on the SSM/I–SSMIS series.

We will further clarify this decision under section 3.2.1.

DONE

**Lines 79–80:**
This line appears to be misplaced. It discusses active microwave (backscatter), whereas the rest of the paragraph focuses on PMW (brightness temperature/emissivity). For passive microwaves, snow layer emissivity increases, especially at higher frequencies (see Dolant et al., 2016). At lower frequencies, TB may decrease due to loss of soil emission, but with a sufficiently deep snowpack at K-Ka bands, TB increases. In active microwave observations, volume scattering is lost with increasing snow depth, leaving only surface scattering. Much of the signal is scattered away from the radar sensor, resulting in lower backscatter (see Tedesco et al., 2015).

We appreciate the reviewer's observation and agree that the referenced sentence introduces active microwave terminology and principles, which is inconsistent with our study. In the revised manuscript, we will clarify this by rephrasing the sentence as follows:

'In contrast, as the liquid water content (LWC) within the snowpack increases, snow emissivity at lower frequencies likely decreases due to attenuated soil emission, while increasing at higher frequencies such as K and Ka bands due to enhanced emission from the wet snow layers (Dolant et al., 2016).'

DONE

**Lines 89–91:**
Consider including the review by Meloche et al. (2022) in this section. They also used a Random Forest model to retrieve snow depth, and their findings are relevant to this study.

We will add Meloche et al. 2022

DONE

**Lines 311–312:**
Please elaborate on how the 80/20 data split was performed. Was the split randomized? If not, correlated data may have affected model performance.

The 80/20 split was random and performed with replacement. The revised sentence will read as follows:

'In each iteration, a random 80/20 (training/testing) split with replacement ensured that the testing data represented a unique subset of high-quality observations from different years and locations, allowing the model's generalizability to be evaluated across a variety of conditions.'

DONE

**Lines 323–325:**
Discuss potential errors arising from differences in spatial scale among datasets. The RF model outputs are at kilometer scale, while GHCNd station data is point-based and may not capture broader spatial variability.

This spatial discrepancy is discussed at line 491-494. Here we discuss the limitations in point-based observations not fully capturing the broader spatial variability.

DONE

**Lines 383–384:**
Consider adding a short paragraph confirming variability in model outputs. Since the models are applied to different time frames, snow cover may be more stable during the IMS period than the SnowModel period. Are the histograms of both models similar, or does the SnowModel histogram show more spread? This may influence interpretation and RF model precision.

This is a great point. We will add a few sentences describing the variability in model outputs and include histograms in the Appendix. Reviewer 2 had similar comments.

DONE

**Line 389:**
Why are 7–10 days not categorized? Is this due to a lack of data in that range?

We apologize, rather than greater than 10 days, it should day 'greater than 7 days.' Thank you making this observation.

DONE

**Lines 398–400:**
Why do the temporal categories (number of days) differ between the two predictions?

The categories are based off a natural breaks classification which differed between snowmelt onset and snowoff. We would like to keep the two predictions in their respective categories and will add the following text to add clarity:

'Given that snow cover at higher elevations can persist for extended periods of time and through the summer months, it is not a surprise that these pixels are ranked as 'low.' The temporal error categories for snowoff differ slightly from those used for snowmelt onset due to differences in the timing dynamics of the two events—snowoff tends to occur more gradually and can span longer periods, especially in complex terrain, necessitating broader classification thresholds to reflect the distribution of prediction errors.'

DONE

**Lines 403–405:**

See earlier comment regarding TB measurement uncertainties in high-topography areas.

We will add to the discussion on the potential impacts in TB measurements in topographically complex areas.

**Section 5.4:**

Please clarify why the time series was split at 2006. The rationale is currently unclear.

Thank you for this comment. We split the time series at 2006 to divide the 36-year period (1988–2023) into two equal halves in order to examine whether any meaningful trends were present within sub-periods of the record. The rationale for this segmentation was that, although the full-period trends were generally nominal or non-significant, we hypothesized that shifts in snow phenology may have occurred within shorter timeframes that would be obscured when analyzing the entire dataset at once. By splitting the record evenly, we aimed to identify any potential intra-period changes or reversals in trend direction, which indeed became evident in the contrasting patterns observed before and after 2006.

We will add the following paragraph under section 4.2.1:

'Because trends over the full period of record were generally weak or non-significant, we further examined the potential for meaningful sub-period patterns by dividing the 36-year time series (1988–2023) into two equal halves: 1988–2005 and 2006–2023. This split allowed us to evaluate whether changes in snow phenology were occurring within shorter timeframes that may have been masked by variability across the full record. Indeed, this approach revealed distinct trends within each period that were not apparent in the overall time series.'

DONE

**Line 505–506:**

This observation may also result from higher TB measurement errors in high-altitude regions.

Here at 505-506 we will discuss implications of higher TB measurements at higher altitudes.

DONE

**Equations:**

Equations throughout the manuscript are not numbered. Please revise to include proper numbering.

We will number the equations.

DONE

References:

Dolant, C., Langlois, A., Montpetit, B., Brucker, L., Roy, A., & Royer, A. (2016). Development of a rain-on-snow detection algorithm using passive microwave radiometry. *Hydrological Processes, 30*(18), 3184–3196. https://doi.org/10.1002/hyp.10828

Forman, B. A., & Xue, Y. (2017). Machine learning predictions of passive microwave brightness temperature over snow-covered land using the Special Sensor Microwave Imager (SSM/I). *Physical Geography, 38*(2), 176–196. https://doi.org/10.1080/02723646.2016.1236606

Liebe, H. (1989). MPM—An atmospheric millimeter-wave propagation model. *International Journal of Infrared and Millimeter Waves, 10*(6), 631–650. https://doi.org/10.1007/BF01009565

Meloche, J., Langlois, A., Rutter, N., Royer, A., King, J., Walker, B., Marsh, P., & Wilcox, E. (2022). Characterizing tundra snow sub-pixel variability to improve brightness temperature estimation in satellite SWE retrievals. *The Cryosphere, 16*(1), 87–101. https://doi.org/10.5194/tc-16-87-2022

Tedesco, M., Sartori, M., & Jeyaratnam, J. (2015). An overview of the current NASA operational AMSR-E/AMSR2 snow science team activities. In *2015 IEEE International Geoscience and Remote Sensing Symposium (IGARSS)* (pp. 4037–4040). https://doi.org/10.1109/IGARSS.2015.7326711

Xiao, X., Liang, S., He, T., Wu, D., Pei, C., & Gong, J. (2021). Estimating fractional snow cover from passive microwave brightness temperature data using MODIS snow cover product over North America. *The Cryosphere, 15*(2), 835–861. https://doi.org/10.5194/tc-15-835-2021

Xiong, C., Yang, J., Pan, J., Lei, Y., & Shi, J. (2022). Mountain snow depth retrieval from optical and passive microwave remote sensing using machine learning. *IEEE Geoscience and Remote Sensing Letters, 19*, 1–5. https://doi.org/10.1109/LGRS.2022.3226204

Response to Reviewer 2:

We thank Dr. Dunmire for taking the time to read our manuscript and provide a constructive and thorough review. We did our best to address each comment and believe the revised manuscript will be greatly improved. Our responses to each comment were highlighted in pink.

**Major Comments**

My major concern with this work is that it seems that the RF approach isn't really necessary. In fact, for snowmelt onset, the previous thresholding approach appears to do better (MAE of 11 days compared with 11.6 days for RF), and the IMS dataset does better for snowoff timing (MAE of 15.87 days compared to 16.03 days for random forest). Why can a combination of the thresholding approach for snowmelt onset, and IMS for snowoff not be used then?

Thank you for this important question. While it is true that the MAE for snowmelt onset in the Pan et al. (2020) threshold-based approach was slightly lower (11.0 vs. 11.6 days), the RF model offers several advantages not reflected in these summary metrics:

1. The thresholding approach tends to misclassify temporary warming events as snowmelt onset, particularly in early spring. The RF model better distinguishes transient melt events from true snowmelt onset by incorporating time-series context and multiple predictors (e.g., TDD, snow cover, DOY). As summaried in our manuscript (Conclusion section), "One significant advantage of the RF approach over traditional thresholding methods is its reduced sensitivity to transient melt events and atmospheric fluctuations, making it more

reliable in identifying primary snowmelt onset rather than temporary thaw and early melt events caused by brief warming episodes".

2. The RF model performs better in regions with complex terrain, high fractional water, or inconsistent snowpack dynamics, where the thresholding approach tends to be oversimplified under the complex conditions. This is evident in the QAQC maps and error spatial distributions we present.

3. The RF framework is more flexible and capable of integrating additional data sources and predictors (e.g., SAR), enabling continuous model evolution and refinement.

4. The RF or alternative machine learning methods can be adapted to other regions including other large northern hemisphere basins, while the performance of thresholding methods may vary with locations.

5. As shown in our bootstrapping uncertainty analysis, the RF model achieved mean absolute errors of 5.8 days for snowmelt onset and 5.0 days for snowoff when the testing data closely matched the distribution of the training features. This suggests that, under well-matched conditions, the RF model can achieve substantially lower errors than the average MAE values reported across the full domain.

Further, in the conclusion, it is stated that this study delivers "new insights into the timing and variability of snow melt onset and snowoff," but I actually fail to see these new insights. A lot of this phenology was already presented in Pan et al. 2021 (i.e., section 3.2). Is there actually any new insights that are gained from the RF approach, that are absent when using the thresholding method?

We appreciate the opportunity to clarify this point. While this study was built on Pan et al. (2020), it represents a more comprehensive assessment of snow phenology and driving factors over an extended period as detailed below:

This study extends the snow phenology record to 2023, adding five critical years during which rapid Arctic warming and record-breaking temperatures occurred.  The long-term record enables the identification of the accelerated changes in snow phenology in recent years (2006-2023). The findings were summarized in the Conclusion as "Finally, this study produced an extended snow phenology record spanning more than 30-years to better distinguish climate normals and quantify long-term climate trends in the YRB... However, in more recent years (2006–2023), both snowmelt onset and snowoff timing have advanced significantly, coinciding with rising spring and summer temperatures across the YRB. These phenological shifts, along with the lengthening of the snow-free season, align with observed patterns of earlier spring onset and more frequent anomalous warming events in recent years."

In addition, the RF model predicts daily snow conditions (wet/dry, present/absent), enabling more detailed description and finer analysis of the melt season's evolution than the thresholding approach (Pan et al., 2020).

We also evaluated model uncertainty over heterogenous surface conditions (FW, elevation, proximity), and produced spatial error maps, which provided improved quantification of model

uncertainty and detailed interpretation of spatial heterogeneity impacts on passive microwave snow retrievals.

The motivation for this work needs to be substantially improved. Given the previous work from 2020, which already presents a 30-year snow phenology record of the YRB with many of the same insights, what does this work add? Why is current knowledge of snow-melt and snowoff conditions in YRB insufficient?

As suggested, we will better clarify the motivation of this work in the Introduction section as follows:

"While threshold-based methods have successfully predicted snow phenology, they often fail to fully capture landscape variability in snow conditions due to their coarse spatial resolution"

"While prior studies, including Pan et al. (2021), provided valuable long-term snow phenology records using threshold-based approaches, these methods are often sensitive to transient melt events, rely on static thresholds, and lack spatially explicit uncertainty estimates. Additionally, previous datasets either span shorter time periods or use lower-resolution inputs, limiting their ability to resolve snowpack dynamics in heterogeneous terrain. To address these limitations, we develop a Random Forest-based approach that integrates dynamic satellite indices and static landscape features to produce a consistent, high-resolution, and uncertainty-informed snow phenology record from 1988 to 2023".

For additional clarity Pan et al. 2021 presents a 28-year record (1988-2016). The data record was extended to 2018 after publication.

Second, some details are lacking about the data/methodology. Also, in general, a schematic of the workflow would be incredibly helpful. Please see the below comments, organized by section:

With the revision we feel we have provided enough detail and description such that a workflow diagram is not needed.

DONE

**Section 3.2.1**
L165 – How frequent are missing temporal observations in this data?

Thank you for your comment. Most data gaps were resulted from short-term sensor outages, orbital gaps, or data acquisition issues, which are generally limited to 1–2 consecutive days (e.g., Wang et al., 2016; Long and Brodzik, 2016).

We will update our paragraph to the following, update in bold:

**3.2.1 Passive Microwave Satellite Record**
We acquired K-band (19 GHz) and Ka-band (37 GHz) afternoon Tb retrievals at vertical (V) and horizontal (H) polarizations from the MEaSUREs Calibrated Enhanced Resolution Passive Microwave Daily EASE-Grid 2.0 Brightness Temperature ESDR, available from the National Snow and Ice Data Center (NSIDC) (Brodzik and Long, 2016). This Tb record is multidecadal and calibrated across multiple sensors and platforms from different frequencies and polarizations from the NOAA DMSP SSM/I and SSMIS. Each platform has several sensors; from SSM/I we selected F08 (1988–1991), F11 (1992–1995), and F13 (1996–2007), and from SSMIS we used F17 (2007–2016) and F18 (2017–2023). These sensors were selected because their equatorial overpass time remained consistent while in commission. Missing temporal observations were gap-filled using a temporal linear interpolation of adjacent Tb retrievals (Wang et al., 2016). **These missing observations are generally infrequent and short in duration, typically affecting only 1–2 consecutive days due to sensor outages, orbital gaps, or data transmission gaps (Long and Brodzik, 2016; Wang et al., 2016).**

DONE

**Section 3.2.2**
Isn't there also a 1km IMS product that exists? Why was this not used?

Yes, IMS 1km record is available from 2014-2025 (https://noaadata.apps.nsidc.org/NOAA/G02156/1km/) . We chose not to use the 1km because of its limited temporal coverage.

DONE

I also find details about SnowModel to be lacking here. What is the model forced with?

SnowModel is forced with MERRA-2 and ERA5 as well as NSIDC sea ice parcel concentration and trajectory datasets (Liston et al. 2020). We will add this description to the manuscript text.

DONE

Is it consistent with IMS when the two products overlap?

Yes, where data overlap, SnowModel and IMS are consistent and possess similar results. SnowModel and IMS mean and median snowoff values are often within one day. We will add this figure to the Appendix. Reviewer 1 had a similar suggestion as well.

[Figure]

DONE

**Section 3.2.3**
How was the cumulative thaw degree days calculated? Cumulative from what point in time?

Thank you for this comment. We will include the following in the revised manuscript:

"TDD was created by summing daily mean temperatures above 0°C for each and is returned as a cumulative percent."

DONE

**Section 4.1**
Did you try any other ML models such as xgboost or neural networks?

We tested a Long Short Term Memory approach, which did not outperform the RF method. We will discuss the use of alternative machine-learning methods, in particular boosted trees and neural networks in the revised manuscript.

DONE

**Section 4.1.1**
How many combinations of parameters did you try in the grid search?
What were the parameters that were changed and what was the range of values tested?
Please also reference Table A2 in this section.

We updated TableA2 to include which parameters we tried in the grid search. We will also add the following to the text: 'A full list of tested parameter ranges and the selected final values for each model are provided in Table A2.'

|  | Values Tested | Snowmelt Onset | Snowoff |
|---|---|---|---|
| n_estimators | 8, 100, 200, 300, 1000 | 100 | 100 |
| min_samples_leaf | 1, 2, 4 | 4 | 4 |
| max_depth | None, 10, 20, 30 | 10 | 10 |
| max_features | auto, sqrt, log | log2 | log2 |
| min_samples_split | 2, 5, 10 | 5 | 10 |

DONE

**Section 4.1.2**
More details in this section would be helpful. For example:

How was the split between training and testing done? How was the feature importance computed? Why was the QC metric split into 4 categoric categories, instead of just kept as a number between 0 and 1? Is there information that is lost in doing this?

Thank you for this comment we will update 4.1.2 as follows, additions in bold:

Model performance was assessed using a bootstrapping approach, with an 80/20 split between training and testing data, with replacement, allowing us to evaluate model accuracy and variability. **For each iteration, the training and testing datasets were randomly sampled with replacement from the full dataset, stratified to ensure balanced representation across land cover types.** Performance was evaluated with the training data by (1) extracting the $R^2$ value to quantify the agreement between observed and predicted dates, and (2) aggregating the Mean Absolute Error (MAE) across different land cover types. To determine whether differences in model error across land cover characteristics were statistically significant, we applied a one-way Analysis of Variance (ANOVA).

**Feature importance was computed using the built-in mean decrease in impurity (MDI) method from the Random Forest algorithm. This approach evaluates the contribution of each feature to the reduction in overall model error across all trees, averaged across iterations. We then calculated the mean and standard deviation of each feature's importance across all bootstrap runs to assess consistency.**

The output absolute error from our model bootstrapping was used as the dependent variable in an ordinary least squares (OLS) regression, with land cover variables such as FW, TC, elevation, aspect, and proximity serving as explanatory variables (Kim et al., 2011). The goal was to establish a relationship between the observed error and the land cover characteristics to identify pixels in the YRB where we may expect lower or higher errors. We then applied the OLS model across the YRB to predict anticipated error. These values were scaled from 0 to 1, creating a dimensionless quality control (QC) metric.

**The continuous QC values (0 to 1) were further classified using a natural breaks (Jenks) classification to define four relative quality categories: 'Best,' 'Good,' 'Moderate,' and 'Low.'** This qualitative classification simplifies interpretation and communication of model quality across the basin, especially in map products or stakeholder-focused outputs. **While this process discretizes a continuous variable, the underlying QC values are retained and available for quantitative analysis.**

DONE

**Section 4.2.1**
The classification into 'earlier' and 'later' snow events is not clear. What does 'snow event' mean specifically? Why was this done?

Thank you for the helpful comment. We will update section 4.2.1 as follows:

In this context, a "snow event" refers to an annual snow phenology transition—such as snowmelt onset or complete snow disappearance—at the pixel level. Each pixel's event date was compared to its long-term climatological mean (1991–2020). If the event occurred earlier than the mean, it was classified as "earlier"; if later, it was classified as "later."

This classification enabled us to calculate the total area (in km²) of the basin experiencing earlier or later snow events in each year. This spatially aggregated approach provides a more hydrologically meaningful metric than raw day-of-year values, as it reflects how much of the landscape is contributing to earlier or delayed runoff. Framing the results relative to the climatological mean also contextualizes each year's snowmelt timing within a long-term baseline, helping to interpret the magnitude and direction of interannual variability.

DONE

**Section 4.2.2**
Please specify what this data was used for.

We will update this section as follows, with additions in bold:

Seasonal air temperature across the YRB was analyzed using data from GHCNd climate stations **to assess long-term climate trends and their relationship to changes in snowpack conditions.** To create a single, harmonized air temperature time series, we selected stations with at least 17 years of data for each of the two time periods (1988–2005 and 2006–2023) from an initial set of 35 stations in the YRB. This selection criterion reduced the set to 8 stations for 1988–2005 and 15 stations for 2006–2023. With the selected stations, we then calculated seasonal average temperature time series for each period, specifically for winter, spring, summer, and combined spring/summer temperatures. **These seasonal means were used in a trend analysis**

**to evaluate how air temperatures have changed over time across the YRB and to explore their potential associations with observed snowmelt dynamics.**

We also extracted the snow depth at the day of snowmelt onset across YRB using the GHCNd climate stations. Like temperature, we required a station to have recorded at least 17 years of snow depth. We also checked each of these stations to determine if the annual snow depth measurements were complete, because they are often incomplete. These screening criteria resulted in 4 stations selected for 1988–2005 and 13 stations for 2006–2023. **The snow depth data were used to analyze trends in snow**

DONE

**Minor/Technical Comments:**

L66: "While higher-frequency K-band…" Perhaps this line should go to the next paragraph? It seems a bit out of place in this paragraph which primarily covers C-band SAR and Sentinel-1.

Agreed. Reviewer 1 had a similar response. We will remove this sentence and place it at the end of the previous paragraph.

DONE

Figure 1 – Please change the triangle markers to dots.

We will change the triangle markers to dots.

DONE

Section 3.1 – This study is for the YRB, but many of the sites used are located outside the YRB, particularly a large number along the southern coast. I understand this is likely done to enhance the limited in-situ data available to train the model, but the impact of this should be mentioned somewhere.

We will add the following the section:

While our focus is on the YRB, we included sites located outside the basin to supplement the sparse in-situ observations within the YRB and enhance model generalizability. This expanded dataset increases the robustness of the model across diverse snowmelt regimes. However, we acknowledge that incorporating data from regions with different climatic conditions (e.g., maritime vs. continental) may introduce some bias, and we interpret model outputs within the YRB with that potential limitation in mind.

DONE

L140 – What is meant by 'analysts'?

We will change 'analysts' to 'we'.

DONE

L144 – Why is there a different number of observations available for training snow melt onset and snowoff?

There are certain years for a given station where snow depth measurements end before the last day of snow and so in those circumstances, we cannot assess a snowoff observation. We will clarify this in the revised manuscript.

DONE

L149 – Please specify that the snow cover is binary.

We will specify snow cover is binary.

DONE

L146–155 – Please mention somewhere in these paragraphs that these variables will be described in more detail in the following sections.

Good point. We will add this clarification in the revised manuscript.

DONE

L153 – "GTOPO" in parenthesis is not needed.

We will remove 'GTOPO.'

DONE

L172–174 – Please change "reduce" to "reduced" in these sections to be consistent with the tense throughout the methods section. Check other areas in the methods as well.

Thank you for this comment. We will change 'reduce' to 'reduced' and check for incorrect tense.

DONE

L176, L178 – The equations are numbered in the text, but not in-line. If possible, please include a number next to the equations for easy reference.

We will number equations in-line.

DONE

L209 – Can the 'TC' abbreviation be changed to 'FTC' to be consistent with 'FW' ("fractional water")?

Absolutely, we will change TC to FTC.

DONE

L226 – I do not fully understand the sentence: "Although RF does not inherently model temporal sequences like some other algorithms, temporal dimensions were incorporated by structuring each day as a sample within a sequential framework."
Specifically, the 'within a sequential framework' is throwing me off. RFs don't have a sequential framework, and each sample is treated distinctly, not as part of a sequence. Can you please clarify this here?

Thank you for pointing this out. Unlike Recurrent Neural Networks designed for processing time series data, the RF model is not able to capture temporal patterns automatically. In our study, while the RF daily inputs are treated independently by the model, the resulting daily predictions are interpreted sequentially to determine transition points such as snowmelt onset or snowoff. This sequencing allows us to extract accurate phenology metrics from the model outputs.

We will update the paragraph accordingly, updates in bold:

We implemented a RF classifier to predict snow phenology, specifically focusing on estimating the annual timing (day of year) of snowmelt onset and snowoff across the YRB. The RF approach was chosen due to its ability to handle complex, high-dimensional data, and robustness to overfitting, making it well-suited for cryosphere applications (Breiman 2001, Alifu et al., 2020; Blandini et al., 2023). Although RF does not inherently model temporal sequences like some other algorithms, **our approach uses daily inputs to generate a sequence of predictions that are later interpreted in chronological order**. Each day is treated as an independent sample during training and prediction, but **the outputs are interpreted sequentially to identify the timing of snowpack transitions**, such as the first wet snow day indicating snowmelt onset. **This post-prediction sequencing is critical for deriving the correct day-of-year phenology metrics.** Accordingly, we used the RF implementation in scikit-learn (Pedregosa et al., 2011).

DONE

L230 – What were the spatial and temporal resolution of the other similar snow record? Is it really correct to say that this study is a "significant temporal enhancement"?

The previous snow record was at 6.25 km and the spatial resolution of the record presented in this paper is 3.125 km. As for the temporal resolution the threshold based snow records were originally 1988-2016 and those are the years analyzed in Pan et al. 2021. Those data were extended to 2018 and we used the extended version in this paper (1988-2018). With that said, the dataset presented in this paper is 7 years longer than the one previously published. It might not be wrong to say a 'significant temporal enhancement' when the new record is 25% longer than the previous. We will change 'significant' to 'valuable.'

DONE

L239 – "By labeling the timeseries accordingly…" It is not clear to me how labeling the timeseries this way enables any of the following numbered points.

To be clearer, we will also update the following text, updates in bold:

To enhance the prediction of snow phenology, we configured the RF model to delineate daily snow conditions. We did this by classifying expected snow conditions for each day in a time series leading up to the observed snowmelt onset or snowoff day in spring as either 'dry snow' or 'present'. After the observed onset or snowoff day, the conditions are labeled as 'wet snow' or 'absent', respectively. **This labeling approach transforms each time series into a sequence of daily snow condition classes, enabling a clear and daily description of a given evolving phenological event.** By labeling the time series accordingly, we were able to

DONE

Figure 2 – In the bottom panel, it would be more clear to set the y-axis as merely "dry" for 0, and "wet" for 1.

In line. We will make those changes accordingly.

DONE

L302/306 – Are these the scores of the training dataset? Testing?

Thank you for your comment. We will make the clarification in the text.

DONE

L315 – Is there any bias in the snowmelt/snowoff timing?

Thank you for this comment. We will add bias to denote if there is any bias and in what direction it occurs for the two datasets.

DONE

L320 – What is "the full YRB dataset"?

We will clarify as follows, changes are in bold:

'The final error assigned to the snow phenology dataset is assessed by comparing the RF model outputs **across the full YRB gridded dataset (i.e., model predictions for all pixels across the basin)** with **an independent validation dataset derived from** the limited number of GHCNd stations **located** within the YRB. From these stations, we calculated a MAE of 11.6 days and RMSE of 14.9 days for snowmelt onset. The model's snowoff results showed a MAE of 18.1 days and RMSE of 21.3 days relative to the station observations. **The higher final observed errors, compared to the bootstrapped model errors, are likely attributed to the greater heterogeneity in land cover and terrain across the basin, which is not fully represented by the sparse in-situ station network.**'

DONE

L322–323 – It is not clear to me how these MAE values differ from the ones presented in L315–317.

The first set of MAE are from the boostrapped approach using data inside and outside the YRB while the second using the testing observations within the YRB to calculate MAE on the final model outputs.

We will clarify the paragraph accordingly:

Once the snowmelt onset and snowoff days of year (DOY) were extracted, they were compared against our testing data generated during bootstrap iterations. These stations are located both inside and outside the YRB

The final error assigned to the snow phenology dataset is assessed by comparing the RF model predictions across the full YRB gridded dataset with an independent testing dataset derived from the limited number of GHCNd stations located within the YRB.

DONE

Section 5.1.2 – What are you using as the ground truth in this section to compute the MAE?

Thank you for this comment. We will clarify that the MAE values in this section are derived from bootstrapped model predictions, and that the observed snowmelt onset and snowoff dates

used within the bootstrapping framework serve as the ground truth for computing error by land cover type.

We will adjust 5.1.2 as follows, updates in bold:

**5.1.2 Landcover and Uncertainty**

Mean absolute errors were binned by land cover to assess whether land cover characteristics had a significant influence on model performance. **These MAE values were derived from the bootstrapped model predictions, aggregated across all iterations and grouped by land cover type.** Land cover features such as elevation, TC, aspect, proximity, and FW were grouped into four natural breaks and compared with the corresponding model MAE to identify potential patterns or relationships. Figures 3 and 4 indicate that when elevation, proximity and FW decrease, MAE also decreases for both snowmelt onset and snowoff predictions. Conversely, as TC increases, MAE also increases, though this is only observable for snowmelt onset predictions. Also notable is that higher FW is associated with much higher MAE values for snowmelt onset. Hence, FW may be a major factor behind the overall lower model performance, relative to snowoff. Yet, overall, these landcover and error interactions are as anticipated – error increases with higher surface water cover, coastal proximity, and tree cover.

DONE

L335 – The 'FW' feature was not grouped into 4 natural breaks (not a variable in Fig. 3).

We will clarify why we did not aggregate FW.

FW was not aggregated by natural breaks because there were too few unique values.

DONE

L339 – "Higher FW is associated with much higher MAE values for snowmelt onset."
I don't see this in Figure 4. FW of 10 and 13% have by far the highest MAE and then it decreases for higher values. Maybe I'm misunderstanding this figure?

Thank you for pointing this out. You are correct that Figure 4, as currently presented, does not clearly reflect the intended interpretation. Specifically, FW values of 10% and 13% are relatively high, and the figure is likely skewed by the small number of observations at those levels. In the revision, we will improve Figure 4 by normalizing FW by the number of observations within each FW bin, which will better represent the relationship between FW and model error.

'Model error varied across FW values, with some mid-range FW bins (e.g., 10% and 13%) exhibiting elevated MAE. However, this pattern may reflect sampling artifacts due to fewer observations in these bins, rather than a consistent relationship between FW and error.'

DONE

Figure 3 – I think the division into 'four natural breaks' is not super helpful in this figure.
I guess we can see that MAE for snowmelt is higher at higher elevation, but what are the elevations?
Also, it doesn't make sense for the aspect feature as aspect describes a direction that has a cyclical nature, not really a standard increasing variable.

In line. We will rework this figure to include MAE and the actual land cover value, rather than binning. The intent was to show a more 'relative' perspective.

DONE

Figure 4 – The x-axis for this plot is confusing to me. What is it binned so seemingly randomly? For example, the x-axis jumps from 10 to 11 to 13 to 22.

We addressed this in the earlier comment. But the randomness is because it is not classed on natural breaks and shows the actual unique values. To mitigate confusion we may remove this figure or modify in the revision.

DONE

L341 – Why is it anticipated that error increases with higher surface water cover?

Thank you for this question. Higher FW cover introduces greater uncertainty in the coarse-resolution passive microwave satellite observations due to the large contrast in emissivity between water bodies and the surrounding land features (e.g., snow, soil, and vegetation). Open water has a low emissivity and can mask or distort the snow signal by dampening the contrast between snow-covered and snow-free land.

We will add the following:

Higher error in areas with greater surface water cover (FW) is expected, as open water has low microwave emissivity and can obscure or distort the passive microwave signal used to detect snow state transitions.

DONE

L352 – "These results support inclusion of the static variables as additional RF predictors, despite their relatively low importance." I'm not totally convinced by this. All the ANOVA results are saying is that the MAE is statistically significantly different for different land cover types, not that the land cover type is helping improve the model MAE, right? This analysis could indicate that perhaps your model is missing processes for certain land cover types, thus influencing the MAE for these land cover types, but I don't think it indicates that the static variables are helping

improve the model. Did you test models with these static variables omitted to see how the MAE changes? This would give a better indication that the static variables are useful.

Thank you for this comment. We agree that the ANOVA results indicate differences in model error across land cover types but do not, by themselves, justify inclusion of the static variables in the model. We will revise the text to clarify this distinction and acknowledge that additional testing—such as running the model with and without static variables—would be needed to assess their actual contribution to improving performance.

We will update the text accordingly, updates in bold:

The one-way ANOVA results indicate that each land cover characteristic has a significant (p < 0.0001) influence on MAE. For both snowmelt onset and snowoff, FW had the greatest impact with an F-statistic = 40.63. TC and proximity also showed substantial effects on MAE, with F-statistics of 21.34 and 30.18 for TC, and 10.25 and 19.22 for proximity, for snowmelt onset and snowoff, respectively. **These results indicate that model performance varies significantly by land cover type, suggesting that certain land cover characteristics are associated with higher or lower prediction errors. However, ANOVA alone does not demonstrate whether these static variables improve model performance when included as predictors. Future work could evaluate this more directly by comparing model performance with and without these static variables to determine their contribution to reducing prediction error.**

DONE

Section 5.4.2 – Are these anomalies better captured by the RF or are they already seen in other records?

The anomalous years described in Section 5.4.2 are apparent in the observational snow phenology records derived from satellite and in-situ data, and are not uniquely identified by the RF. However, the RF model does successfully capture the timing and magnitude of these anomalies in its predictions, demonstrating its ability to reproduce interannual variability and respond to extreme climate conditions.

We can add something as follows to the section: 'These anomalies likely reflect broader climatic drivers, such as temperature fluctuations and abnormal precipitation, affecting snowmelt dynamics in these years. While these deviations are visible in the observational record, the RF model successfully reproduced the timing and magnitude of the anomalies, indicating its sensitivity to interannual climate variability.'

DONE

L421 – "Snowmelt onset occurred ~9 days earlier than average, and snowoff 6 days earlier, lengthening snowmelt duration by 4 days…"

Maybe this is a rounding error, but should this not be 3 days? Same with the following sentence. Maybe including decimals would help.

Thank you for this catch. We will change '4' to '3.'

Section 5.5.2 – Can a figure be included for the trends in seasonal temperature? Also, what does the 'tau' value represent?

Yes we will include a figure for trends in seasonal temperature. Tau is a non-parametric rank correlation produced in the Mann-Kendall Test and provides an indicator of the direction and strength of two variables, here it's time and temperatures.

Description added in section 4.2.1.

DONE

A figure comparing the performance of the different methods (IMS, SnowModel, thresholding, RF) for snowmelt and snowoff would be helpful.

Thank you for this comment we will consider a figure or table to make it easier to assess differences in model performance.

DONE

References:

Brodzik, M. J. and Long, D. G.: MEaSUREs Calibrated Enhanced-Resolution Passive Microwave Daily EASE-Grid 2.0 Brightness Temperature ESDR, Version 1, https://doi.org/10.5067/MEASURES/CRYOSPHERE/NSIDC-0630.001, 2016.

Kim, Y., Kimball, J. S., McDonald, K. C., and Glassy, J.: Developing a Global Data Record of Daily Landscape Freeze/Thaw Status Using Satellite Passive Microwave Remote Sensing, IEEE Trans. Geosci. Remote Sensing, 49, 949–960, https://doi.org/10.1109/TGRS.2010.2070515, 2011.

Liston, G. E., Itkin, P., Stroeve, J., Tschudi, M., Stewart, J. S., Pedersen, S. H., Reinking, A. K., and Elder, K.: A Lagrangian Snow-Evolution System for Sea-Ice Applications (SnowModel-LG): Part I—Model Description, JGR Oceans, 125, e2019JC015913, https://doi.org/10.1029/2019JC015913, 2020.

Pan, C. G., Kirchner, P. B., Kimball, J. S., Du, J., and Rawlins, M. A.: Snow Phenology and Hydrologic Timing in the Yukon River Basin, AK, USA, Remote Sensing, 13, 2284, https://doi.org/10.3390/rs13122284, 2021.

Wang, L., Toose, P., Brown, R., and Derksen, C.: Frequency and distribution of winter melt events from passive microwavesatellite data in the pan-Arctic, 1988–2013, The Cryosphere, 10, 2589–2602, https://doi.org/10.5194/tc-10-2589-2016, 2016